# When Preference Labels Fall Short: Aligning Diffusion Models from Real Data

**Weiyan Chen** [1]   **Weijian Deng** [2]   **Yao Xiao** [1]   **Weijie Tu** [3]
**Ziyi Dong** [1]   **Ibrahim Radwan** [4]   **Liang Lin** [1 5]   **Pengxu Wei** [1 5]

## Abstract

Preference alignment aims to guide generative models by learning from comparisons between preferred and non-preferred samples. In practice, most existing approaches rely on preference pairs constructed from model-generated images. Such supervision is inherently relative and can be ambiguous when both samples exhibit artifacts or limited visual quality, making it difficult to infer what constitutes a truly desirable output. In this work, we investigate whether real data can serve as an alternative source of supervision for preference alignment. We adopt a data-centric perspective and study a curation strategy that treats real images as reference points and constructs preference signals by contrasting them with generated or perturbed samples, without requiring manually annotated preference pairs. Through empirical analysis, we show that real-data-based supervision provides effective guidance for aligning diffusion models and achieves performance comparable to existing preference-based methods. Our results suggest that real data offers a practical and complementary source of supervision for preference alignment and highlight directions of label-efficient alignment strategies. Code and models are available at `https://cwyxx.github.io/RealAlign`.

## 1. Introduction

Text-to-image diffusion models (Rombach et al., 2022; Esser et al., 2024; Labs, 2024) are typically trained on large-

scale text–image pairs using a likelihood-based objective, which does not explicitly capture human preferences. To address this, recent works (Wallace et al., 2024; Liu et al., 2025) introduce an alignment stage that fine-tunes diffusion models using preference signals, encouraging outputs that better match human judgments of quality and visual appeal. This alignment bridges the gap between likelihood-based training and subjective human expectations.

Preference alignment is commonly formulated through comparisons between preferred and non-preferred samples. By learning from such relative judgments, models can be guided toward outputs that humans find more appealing or more consistent with the input prompt. In practice, however, most existing approaches construct preference pairs from model-generated images. This design introduces several limitations. Because both preferred and non-preferred samples are generated by the generative models, the supervision signal is inherently constrained by the quality of those generators. Even samples labeled as preferred may contain artifacts, lack realism, or exhibit limited stylistic diversity. Figure 1 illustrates this issue using examples from Pick-a-Pic v2 (Kirstain et al., 2023). Although the images in the leftmost column are preferred over their counterparts, they still show noticeable local artifacts or implausible details, and the preferred examples in the right group exhibit unnatural global color distributions and reduced visual realism. Moreover, preference labels provide only relative supervision. When both images are imperfect, the preferred one may still deviate substantially from a truly desirable output, making it difficult for the model to infer how the generation process should be improved. This limits the effectiveness of preference-based supervision for learning robust and generalizable generation behavior. Collecting large-scale preference annotations further requires substantial human effort, making such supervision costly and difficult to scale across diverse styles, concepts, and failure modes.

These limitations raise a natural question: can preference alignment be achieved without relying on manually annotated preference pairs? In this work, we explore whether real data itself can serve as a reliable source of supervision for preference alignment. Unlike model-generated samples, real images, drawn from existing datasets, implicitly reflect human judgments of visual quality and semantic coherence,

---
[1]School of Computer Science and Engineering, Sun Yat-sen University, Guangzhou, China [2]Tsinghua Shenzhen International Graduate School, Tsinghua University, Shenzhen, China [3]Australian National University, Canberra, Australia [4]University of Canberra, Canberra, Australia [5]Peng Cheng Laboratory, Shenzhen, China. Correspondence to: Pengxu Wei <weipx3@mail.sysu.edu.cn>.

*Proceedings of the 43rd International Conference on Machine Learning*, Seoul, South Korea. PMLR 306, 2026. Copyright 2026 by the author(s).

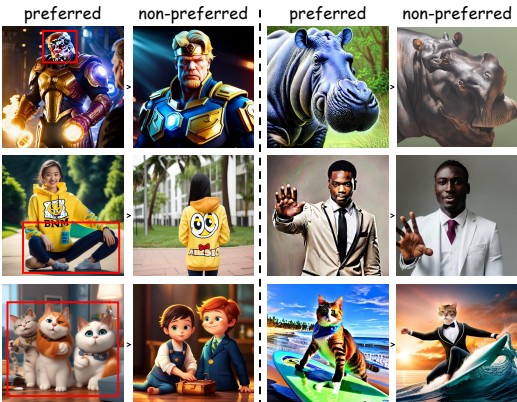

*Figure 1.* **Preference pairs from Pick-a-Pic v2.** The left group shows preferred images with local generation artifacts, while the right presents preferred images with unnatural global color. These cases highlight limitations of preference-based supervision in capturing holistic image quality.

as they are the result of natural selection and curation processes rather than model generation. Compared to synthetic samples, they are less affected by generation artifacts and exhibit broader visual diversity.

Building on this observation, we propose a data-centric formulation of preference alignment, in which real images are used as implicit positive references to guide model alignment. Specifically, we treat human-preferred images as defining a target distribution and construct preference signals by contrasting them with model-generated or locally perturbed samples. In the first stage, real images anchor the model toward high-quality visual distributions. In the second stage, controlled degradations are introduced to generate negative examples that differ primarily along preference-relevant dimensions such as visual fidelity and text–image consistency. This design ensures that the resulting preference pairs emphasize meaningful quality differences rather than incidental variations.

Importantly, this formulation does not rely on explicit human comparisons. Instead, preference signals emerge from deviations between model outputs and real data, allowing alignment to be driven by discrepancies in realism and semantic coherence. In this way, preference learning is reframed as aligning model generations toward the distribution of real data, rather than optimizing relative rankings among imperfect samples. This perspective also naturally complements existing preference-based methods, as it provides an alternative source of supervision that can be integrated without additional annotation cost.

Through extensive experiments, we show that real-data-based supervision can effectively guide diffusion models and achieve performance comparable to conventional preference-based alignment. We evaluate this setting by

post-training Stable Diffusion v1.5 and Stable Diffusion 3.5 Medium using preference signals derived from real images, while keeping the optimization procedure unchanged. With only 512 constructed pairs, the resulting models reach competitive performance across multiple automated metrics. These results indicate that real data can serve as a practical source of alignment signals, reducing reliance on manually annotated preference pairs. In addition, we find this supervision to be complementary to existing alignment methods, providing consistent gains when applied as a post-training step on top of already aligned models. Our contributions are summarized as follows:

- We study preference alignment from a data-centric perspective and show that real images can serve as an effective source of supervision for preference alignment of diffusion models, without relying on manually annotated preference pairs.

- We introduce a simple and effective framework that constructs preference signals by contrasting real images with generated or perturbed data. Through extensive experiments, we show that this real-data-based supervision achieves competitive performance and can be used as a complementary post-training step to existing preference-based alignment methods, yielding consistent improvements.

## 2. Related Work

**Preference Dataset.** HPDv2 (Wu et al., 2023), HPDv3 (Ma et al., 2025), ImageRewardDB (Xu et al., 2023), and Pick-a-Pic (Kirstain et al., 2023) are widely used human-annotated preference datasets, where each example contains of a prompt, two generated images, and a binary preference label. To scale data collection, these images are typically sampled directly from text-to-image models without explicit quality control. As a result, even preferred samples may contain artifacts or limited realism, which can bias the alignment process toward such imperfections. In addition, constructing these datasets requires substantial human annotation effort. These limitations motivate us to explore whether real images can serve as an alternative source of supervision for preference alignment, reducing reliance on manually labeled preference pairs.

**Human-Preference Alignment for Diffusion.** Inspired by the success of reinforcement learning from human feedback (RLHF) in large language models (Ouyang et al., 2022; Rafailov et al., 2023; Shao et al., 2024), recent work has explored aligning text-to-image diffusion models with human preferences. Early approaches (Black et al., 2023; Fan et al., 2023) apply policy gradient methods to diffusion models, but often suffer from scalability issues as the number of prompts increases. Subsequent methods (Xu et al.,

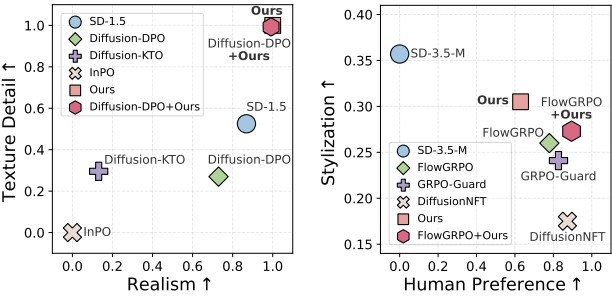

*(a)* Preference Pairs      *(b)* Reward Model

*Figure 2.* **Analysis of preference- and reward-based alignment behaviors.** (a) Comparison of realism and texture detail across different preference-based methods on SD-1.5. Methods optimized using pairwise preferences often improve specific visual aspects (e.g., smoothness or texture consistency) but do not consistently yield balanced gains in overall realism across diverse prompts. (b) Comparison of human preference and stylization on SD-3.5-M. Reward-based approaches achieve higher preference scores but tend to produce more uniform stylistic patterns. (Texture Detail: Laplacian variance of the generated images. Realism: SGP-PickScore (Shen et al., 2025), defined as the difference between PickScore evaluated on prompts prefixed with "Realistic photo" and "CG render". Stylization: stylization score on OneIG-Bench (Chang et al., 2025). Human Preference Score: normalized average of HPSv3 and UnifiedReward evaluated on DrawBench.)

2023; Prabhudesai et al., 2023; Clark et al., 2023; Wu et al., 2024) leverage differentiable reward models to backpropagate through the sampling process, at the cost of substantial memory and computational overhead. Diffusion-DPO (Wallace et al., 2024) avoids explicit reward modeling by directly optimizing pairwise preferences, while later extensions such as FlowGRPO (Liu et al., 2025) and related works (Li et al., 2025; Xue et al., 2025) introduce group-based optimization strategies to improve training stability and efficiency. While these methods focus primarily on algorithmic designs for learning from preference annotations or learned rewards, our work adopts a complementary, data-centric perspective. Instead of introducing new optimization objectives, we investigate whether real images themselves can serve as a reliable source of supervision for preference alignment.

## 3. Preference Signals from Real Data

### 3.1. Real Images as Implicit Preference References

**Potential Limitations of Preference-Pair-Based Supervision.** Most existing preference alignment methods rely on pairwise comparisons between generated samples, where one image is labeled as preferred over another. While this formulation has proven effective in practice, it may introduce several potential limitations.

First, preference pairs are typically constructed from model-generated images. As a result, the supervision signal is constrained by the quality of the underlying generator. Even preferred samples may contain artifacts, limited realism, or stylistic biases, as shown in Figure 1, which restricts the model's ability to learn from truly high-quality references.

Second, preference labels provide only relative judgments between samples. When both candidates deviate from visually desirable outcomes, a preference label does not necessarily indicate what constitutes a high-quality result. In this case, optimization methods such as DPO operate on subtle differences between imperfect samples, which can make it difficult for the model to infer how the generation process should be improved. As illustrated in Figure 2(a), this often leads to biased improvements along specific dimensions, such as increased smoothness or reduced texture variation, rather than a balanced improvement in overall realism.

A similar issue arises in approaches that rely on learned reward models, such as FlowGRPO. Since these reward models are themselves trained on preference data derived from generated images, their supervision reflects the biases present in that data. As shown in Figure 2(b), optimizing against such rewards can improve preference scores while simultaneously encouraging stylistic homogenization or suppressing fine-grained visual details. These observations suggest that both pairwise preference learning and reward-based alignment may be limited by the quality and diversity of the underlying supervision signals.

Finally, collecting large-scale, high-quality preference annotations remains costly and difficult to scale.

**Real Data as a Source of Preference Signals.** Motivated by the limitations of preference supervision based on generated samples, we investigate whether real images can serve as a practical source of supervision for alignment. Real images reflect human choices in visual quality and semantic coherence and are less affected by artifacts introduced during generation. Rather than modifying the preference learning objective, we adopt a data-centric perspective and examine how real images can be used to inform the construction of preference signals. Under this view, real images act as reference examples that capture desirable visual properties, providing an alternative way to guide alignment without relying on manually annotated preference pairs.

### 3.2. Real-Data Curation for Preference Alignment

We present a data curation strategy that constructs structured supervision signals by contrasting real images with controlled variations, without using explicit preference labels. The idea is to first identify a set of images that represent desirable visual properties and then introduce controlled degradations to create informative contrasts. This allows preference-related signals to be derived from real data while keeping the learning process grounded and interpretable.

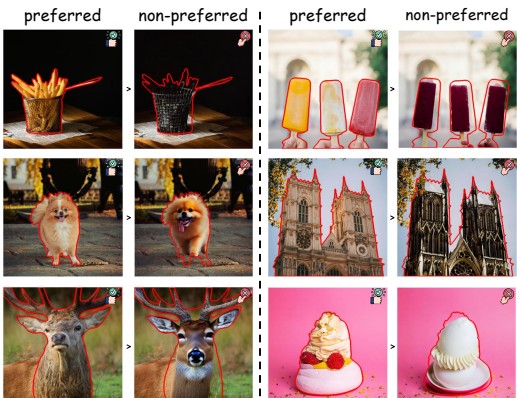

*Figure 3.* **Examples of preference pairs derived from real images.** Red contours indicate salient regions where controlled inpainting introduces localized artifacts. The original images act as preferred references, while the degraded counterparts expose interpretable deviations in texture, structure, or semantics, providing effective supervision for preference alignment without labeling.

**Human-Preferred Images as Reference Samples.** The first step is to construct a set of reference images that reflect desirable visual characteristics. Rather than treating all real images as equally informative, we focus on images that exhibit high visual quality and clear semantic structure. Specifically, we use professional photographs from HPDv3, which contains images curated from photography platforms and filtered based on aesthetic criteria. These images typically exhibit strong composition, coherent content, and high visual fidelity. To further maintain consistency in visual quality, we apply a simple colorfulness-based filter and retain images with above-average scores. This filtering serves as a lightweight heuristic to remove visually flat or low-contrast samples, rather than as a strict measure of aesthetic quality. These curated real images serve as reference samples that define a preferred region of the image space.

**Saliency-Guided Construction of Contrastive Samples.** Preference learning also requires explicit contrasts to identify which deviations should be discouraged. To provide such signals, we construct contrastive samples by introducing controlled degradations to real data. This exposes preference-relevant differences in a localized and interpretable manner, enabling effective learning without relying on manually annotated comparisons.

For each reference image, we first apply a saliency detection model $U^2$-Net (Qin et al., 2020) to identify visually important regions. We then use a prompt-conditioned inpainting model (SD v1.5 Inpainting) to regenerate these regions while preserving the overall layout and semantic structure of the image. Due to the limited expressiveness of the inpainting model, the regenerated regions often exhibit reduced visual quality or weaker alignment with the prompt.

This process produces image pairs in which the original

image serves as a high-quality reference and the inpainted version acts as a degraded counterpart. In practice, we discard cases where the degradation is visually negligible to ensure that the resulting pairs provide a clear and consistent preference signal. The resulting contrasts primarily reflect differences in visual fidelity and semantic coherence, making them suitable for preference-based learning. Figure 3 shows examples of real-image-based preference pairs, where localized and interpretable artifacts provide clear contrastive signals for alignment.

### 3.3. Preference Alignment with Real-Data-Based Signals

Having constructed a reference set that reflects preferred visual characteristics, together with contrastive samples that introduce controlled variations, we next describe how these signals are used in model training. A practical consideration is that real images and their perturbed versions may differ from the model's initial generation distribution, which can make direct preference optimization less stable in practice. To account for this, we adopt a two-stage alignment strategy that incorporates real-data-based signals in a gradual manner. The first stage encourages the model to move closer to the distribution represented by the reference images, while the second stage introduces structured comparisons using the constructed contrastive samples.

**Stage 1: Distribution Alignment with Real Images.** In the first stage, we aim to reduce the gap between the model's generation distribution and the distribution of real images used as reference. Rather than relying on explicit preference comparisons, this stage treats real images as soft anchors that indicate desirable visual characteristics.

We implement this step using a distribution-level alignment objective inspired by inverse reinforcement learning (Abbeel & Ng, 2004). The model is encouraged to assign a higher likelihood to real images than to its own generated samples, while maintaining stability through a reference model. We adopt the optimization objective of Diffusion-DRO (Wu et al., 2025): a reward model $\phi$ is trained to distinguish real images from policy-generated samples in a ranking-based manner; a policy model $\theta$ is updated to reduce this distinguishability, gradually closing the distribution gap between its generated outputs and the real images. The optimal policy model is mathematically identical to the optimal reward model, which allows the alternating optimization between policy learning and reward modeling to reduce to reward modeling alone. This yields the following objective:

$$
\begin{aligned}
\mathcal{L}_{\text{stage-1}}(\boldsymbol{\phi}) = \sum_{t=1}^{T} \mathbb{E}_{\bar{\boldsymbol{x}}_0, \boldsymbol{x}_0, \boldsymbol{c}} \bigg[ \max \bigg( m, \\
- \left( \|\bar{\boldsymbol{\epsilon}} - \boldsymbol{\epsilon}_{\boldsymbol{\theta}_{\text{ref}}}(\bar{\boldsymbol{x}}_t, \boldsymbol{c}, t)\|^2 - \|\bar{\boldsymbol{\epsilon}} - \boldsymbol{\epsilon}_{\boldsymbol{\phi}}(\bar{\boldsymbol{x}}_t, \boldsymbol{c}, t)\|^2 \right) \\
+ \left( \|\boldsymbol{\epsilon} - \boldsymbol{\epsilon}_{\boldsymbol{\theta}_{\text{ref}}}(\boldsymbol{x}_t, \boldsymbol{c}, t)\|^2 - \|\boldsymbol{\epsilon} - \boldsymbol{\epsilon}_{\boldsymbol{\phi}}(\boldsymbol{x}_t, \boldsymbol{c}, t)\|^2 \right) \bigg) \bigg],
\end{aligned}
\tag{1}
$$

where $\bar{\boldsymbol{x}}_t \sim q(\bar{\boldsymbol{x}}_t|\bar{\boldsymbol{x}}_0)$ corresponds to the noisy positive sample derived from the real image $\bar{\boldsymbol{x}}_0$ by adding perturbation noise $\bar{\boldsymbol{\epsilon}} \sim \mathcal{N}(\mathbf{0}, \boldsymbol{I})$; $\boldsymbol{c}$ represents the text prompt condition; and $\boldsymbol{\theta}_{\text{ref}}$ represents the frozen reference model. Conversely, $\boldsymbol{x}_t$ denotes the sample generated from the current policy's distribution, and $\boldsymbol{\epsilon} = \boldsymbol{\epsilon}_{\boldsymbol{\theta}}(\boldsymbol{x}_t, \boldsymbol{c}, t)$ denotes the noise predicted by the policy model. The margin threshold $m$ is introduced to prevent further optimization on already well-ranked samples. This warm-up stage helps bridge the distribution gap between real images and model generations, establishing a good initialization for subsequent pairwise preference learning.

**Stage 2: Preference Learning with Constructed Contrastive Samples.** Once the model has been brought closer to the reference distribution, we introduce preference learning using the contrastive samples constructed in the previous section. These samples are generated by applying controlled perturbations to real images, producing pairs that differ primarily along preference-relevant dimensions such as visual quality and text–image consistency.

At this stage, we apply a standard preference optimization objective, such as DPO, using the constructed pairs. The pairs $(\boldsymbol{x}_0^w, \boldsymbol{x}_0^l)$ are drawn from Section 3.2, where the positive sample $\boldsymbol{x}_0^w$ denotes the real image and the negative sample $\boldsymbol{x}_0^l$ represents its degraded counterpart. Because the positive and negative samples are closely related and differ in targeted ways, the resulting supervision signal is more focused and easier for the model to exploit. This allows the model to refine fine-grained aspects of generation quality without being distracted by unrelated variations. We use Diffusion-DPO (Wallace et al., 2024) as the optimization objective. Both the policy model $\boldsymbol{\theta}$ and the frozen reference model $\boldsymbol{\theta}_{\text{ref}}$ are initialized from the model obtained at the end of Stage 1. The policy model $\boldsymbol{\theta}$ is encouraged to increase the likelihood of the real image $\boldsymbol{x}_0^w$ and decrease that of the degraded counterpart $\boldsymbol{x}_0^l$, while maintaining a KL-divergence constraint against the reference model $\boldsymbol{\theta}_{\text{ref}}$ to prevent distribution shift. The optimization objective is defined as follows:

$$
\begin{aligned}
\mathcal{L}_{\text{stage-2}}(\boldsymbol{\theta}) = -\mathbb{E}_{\boldsymbol{x}_0^w, \boldsymbol{x}_0^l, \boldsymbol{c}} \Big[ \log \sigma \Big( -\beta T \Big( \\
\|\boldsymbol{\epsilon}^w - \boldsymbol{\epsilon}_{\boldsymbol{\theta}}(\boldsymbol{x}_t^w, \boldsymbol{c}, t)\|^2 - \|\boldsymbol{\epsilon}^w - \boldsymbol{\epsilon}_{\boldsymbol{\theta}_{\text{ref}}}(\boldsymbol{x}_t^w, \boldsymbol{c}, t)\|^2 - \\
\|\boldsymbol{\epsilon}^l - \boldsymbol{\epsilon}_{\boldsymbol{\theta}}(\boldsymbol{x}_t^l, \boldsymbol{c}, t)\|^2 + \|\boldsymbol{\epsilon}^l - \boldsymbol{\epsilon}_{\boldsymbol{\theta}_{\text{ref}}}(\boldsymbol{x}_t^l, \boldsymbol{c}, t)\|^2 \Big) \Big) \Big],
\end{aligned}
\tag{2}
$$

where $\boldsymbol{x}_t^w$ and $\boldsymbol{x}_t^l$ represent noisy samples derived from the clean preference pair $(\boldsymbol{x}_0^w, \boldsymbol{x}_0^l)$ via the forward diffusion process at timestep $t$, with $\boldsymbol{\epsilon}^w, \boldsymbol{\epsilon}^l \sim \mathcal{N}(\mathbf{0}, \boldsymbol{I})$ denoting the corresponding ground truth noise. $\sigma$ is the sigmoid function that ensures a proper probability score. The hyperparameter $\beta$ controls the regularization strength, and $T$ denotes the total number of diffusion timesteps.

# 4. Experiments

## 4.1. Experimental Setup

**Datasets and Models.** We construct a paired preference dataset from professional real-world photographs in HPDv3 (Ma et al., 2025) and fine-tune Stable Diffusion v1.5 (SD-1.5) (Rombach et al., 2022) and Stable Diffusion 3.5 Medium (SD-3.5-M) (Esser et al., 2024) using LoRA (Hu et al., 2022). We select the top-512 preference pairs based on the PickScore (Kirstain et al., 2023) of the positive samples. An ablation study on the effect of training data volume is presented in Section 4.4.

**Evaluation.** We generate images using standard prompt sets adopted in prior work. Models based on SD-1.5 are evaluated on the Pick-a-Pic v2 test set (Kirstain et al., 2023), while methods based on SD-3.5-M are evaluated on DrawBench (Saharia et al., 2022), which is consistent with FlowGRPO (Liu et al., 2025). In addition, to examine whether the learned preference signals generalize across different types of prompts, we also report results on Parti-Prompts (Yu et al., 2022), a diverse prompt set covering a wide range of compositional and semantic scenarios. We assess the generated images using six automated metrics. Specifically, we use the LAION aesthetic classifier (Schuhmann et al., 2022) to evaluate aesthetic appeal and DeQA (You et al., 2025) to measure perceptual image quality. To approximate human preference judgments, we report PickScore (Kirstain et al., 2023), ImageReward (Xu et al., 2023), UnifiedReward (Wang et al., 2025), and HPSv3 (Ma et al., 2025). Additionally, we conduct a user study with 18 participants to evaluate our fine-tuned SD-3.5-M model. Detailed settings of this study are provided in Appendix A.3.

## 4.2. Effectiveness of Real-Data-Based Preference Signals

To evaluate whether real images can serve as an effective source of supervision for preference alignment, we first validate this idea on SD-1.5 using preference pairs constructed from professional real-world photographs. As shown in Table 1a, fine-tuning with real-data-based preference signals consistently improves performance over the base model across all evaluation aspects. Despite using only 512 constructed preference pairs, our results are competitive with Diffusion-DPO, which relies on a substantially larger set of manually annotated preference data.

We further assess the effectiveness of real-data-based supervision on the larger SD-3.5-M model to evaluate scalability. As reported in Table 1a, we observe a similar trend: the model fine-tuned with real-data preference pairs yields consistent improvements across all metrics relative to the base model and achieves results comparable to Diffusion-DPO.

To examine whether the preference knowledge learned from

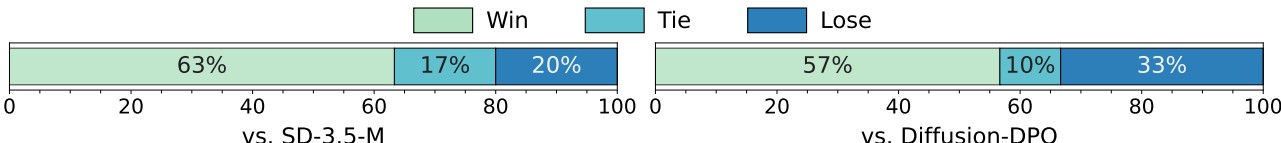

*Figure 4.* **User study on SD-3.5-M.** Following the protocol of Diffusion-DRO (Wu et al., 2025), we randomly sample 60 prompts from HPDv2 and ask users to compare our fine-tuned SD-3.5-M with baselines. Across both comparisons, users prefer our model.

*Table 1.* **Method comparison.** Relative improvements are highlighted in (+gain). ImgRwd: ImageReward (Xu et al., 2023); UniRwd: UnifiedReward (Wang et al., 2025); Aes: LAION aesthetic classifier (Schuhmann et al., 2022). † denotes our re-implementation, trained on the Pick-a-Pic v2 (Kirstain et al., 2023) using official code. Best and second-best results are in **bold** and underlined, respectively.

*(a)* Quantitative results on Pick-a-Pic v2 test set (Kirstain et al., 2023) (SD-1.5) and DrawBench (Saharia et al., 2022) (SD-3.5-M).

| # | Method | Train Data | Preference Score | | | | Image Quality | Aesthetic |
|---|--------|-----------|------------------|--|--|--|---------------|-----------|
| | | | PickScore ↑ | ImgRwd ↑ | UniRwd ↑ | HPSv3 ↑ | DeQA ↑ | Aes ↑ |
| 1 | SD-1.5 | / | 20.65 | 0.16 | 2.92 | 5.98 | 3.70 | 5.48 |
| 2 | Diffusion-DPO | 851k pairs | 21.03 (+0.38) | 0.33 (+0.17) | 3.03 (+0.11) | 6.80 (+0.82) | 3.78 (+0.08) | 5.59 (+0.11) |
| 3 | Ours | 512 pairs | **21.04** (+0.39) | **0.38** (+0.22) | **3.06** (+0.14) | **7.33** (+1.35) | **3.96** (+0.26) | **5.64** (+0.16) |
| 4 | SD-3.5-M | / | 22.42 | 0.79 | 3.30 | 10.03 | 4.09 | 5.44 |
| 5 | Diffusion-DPO† | 851k pairs | 22.70 (+0.28) | 0.97 (+0.18) | **3.47** (+0.17) | 10.79 (+0.76) | 3.96 (-0.13) | 5.44 (+0.00) |
| 6 | Ours | 512 pairs | **22.80** (+0.38) | **1.08** (+0.29) | 3.46 (+0.16) | **12.77** (+2.74) | **4.26** (+0.17) | **5.55** (+0.11) |

*(b)* Quantitative results on Parti-Prompts (Yu et al., 2022).

| # | Method | Train Data | Preference Score | | | | Image Quality | Aesthetic |
|---|--------|-----------|------------------|--|--|--|---------------|-----------|
| | | | PickScore ↑ | ImgRwd ↑ | UniRwd ↑ | HPSv3 ↑ | DeQA ↑ | Aes ↑ |
| 1 | SD-1.5 | / | 21.34 | 0.25 | 3.15 | 5.69 | 3.70 | 5.39 |
| 2 | Diffusion-DPO | 851k pairs | 21.58 (+0.24) | 0.40 (+0.15) | 3.25 (+0.10) | 6.48 (+0.79) | 3.78 (+0.08) | 5.47 (+0.08) |
| 3 | Ours | 512 pairs | **21.64** (+0.30) | **0.41** (+0.16) | **3.28** (+0.13) | **7.06** (+1.37) | **3.96** (+0.26) | **5.49** (+0.10) |
| 4 | SD-3.5-M | / | 22.54 | 1.11 | 3.88 | 8.97 | 4.00 | 5.60 |
| 5 | Diffusion-DPO† | 851k pairs | 22.75 (+0.21) | 1.22 (+0.11) | 3.98 (+0.10) | 9.41 (+0.44) | 3.88 (-0.12) | 5.62 (+0.02) |
| 6 | Ours | 512 pairs | **22.90** (+0.36) | **1.27** (+0.16) | **3.99** (+0.11) | **10.66** (+1.69) | **4.20** (+0.20) | **5.73** (+0.13) |

real data generalizes to different prompt types, we evaluate the aligned models on the diverse prompt set Parti-Prompts (Yu et al., 2022). As shown in Table 1b, we observed consistent improvements over the base model across all reported metrics. Based on these experimental results, real images are shown to be an effective and practical source of supervision for preference alignment in diffusion models.

To complement the automated metrics, we conduct a human evaluation. As depicted in Figure 4, evaluators prefer our fine-tuned SD-3.5-M model over the baselines. These findings align with the automated metric results, confirming that real-data-based supervision provides effective guidance for aligning diffusion models with human preferences.

### 4.3. Complementarity with Existing Preference Alignment Methods

Beyond evaluating real-data-based supervision in isolation, we further examine how it interacts with existing preference alignment methods. Specifically, we study whether incorpo-

rating real-data-derived signals can provide complementary benefits when applied on top of established preference- or reward-based training pipelines. We first apply real-data-based preference signals as an additional post-training stage on top of SD-1.5 aligned using Diffusion-DPO. As shown in the top row of Figure 5, this extra post-training step yields consistent improvements across all evaluation metrics relative to the Diffusion-DPO baseline. The resulting performance is comparable to that of SPO (Liang et al., 2025), a dedicated aesthetic alignment method.

To further illustrate the effect of this complementary stage, we present qualitative comparisons in Figure 6. Models incorporating real-data-derived preference pairs generate images with improved realism and richer texture details, alleviating the biases discussed in Figure 2a. This trend is consistently observed in the larger SD-3.5-M model. Using the representative FlowGRPO as the baseline, post-training with real-data-based preference pairs leads to additional gains across all evaluated aspects, as shown in the bottom row of Figure 5. Qualitative results are presented in Fig-

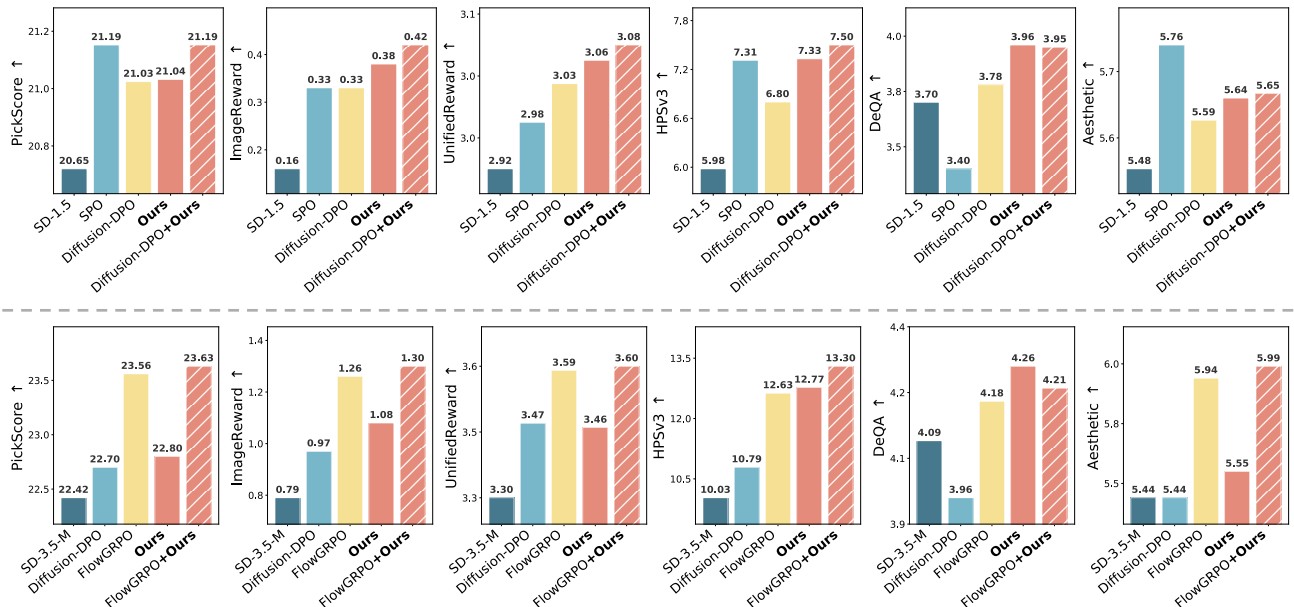

*Figure 5.* **Complementarity with existing preference alignment models.** Top row: Quantitative results on Pick-a-Pic v2 using SD-1.5 as the base model. Real-data-based supervision is integrated with Diffusion-DPO. Bottom row: Quantitative results on DrawBench using SD-3.5-M as the base model. Real-data-based supervision is used as a complementary post-training step on top of FlowGRPO.

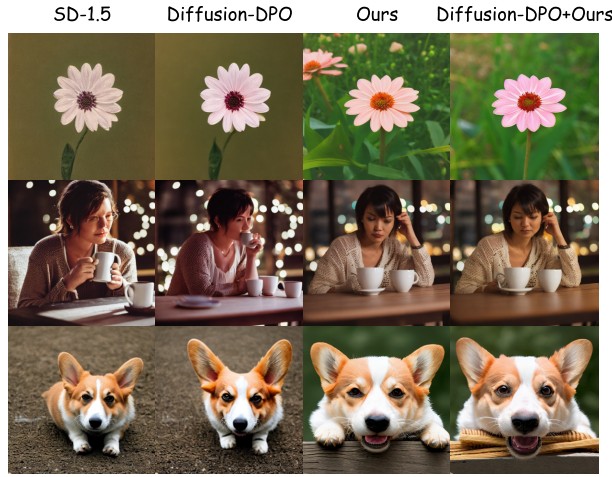

*Figure 6.* **Qualitative comparison based on SD-1.5.** Post-training with real data produces images with improved visual realism and richer texture details. When applied as an additional post-training stage on top of Diffusion-DPO (Wallace et al., 2024), it also improves the visual realism of the resulting generations. Prompts from top to bottom : (1) *a plant.* (2) *a woman sitting on a table drinking coffee, long shot, wide shot, highly detailed, intricate, professional photography, RAW color, night shot, bokeh, sharp focus, taken with EOS 5D, UHD 8K.* (3) *a corgi's head.*

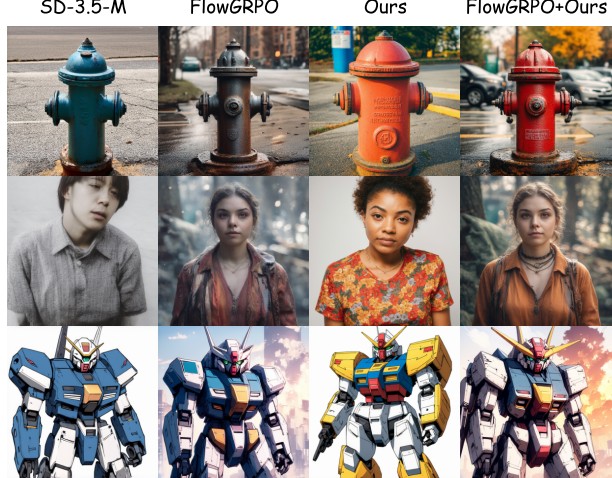

*Figure 7.* **Qualitative comparison based on SD-3.5-M.** Compared to FlowGRPO (Liu et al., 2025), post-training with real data yields more realistic lighting and more natural color distributions. When applied as an additional post-training step on top of FlowGRPO, it further alleviates stylistic homogenization. Prompts from top to bottom : (1) *a fire hydrant.* (2) *a person.* (3) *Anime illustration of Gundam mech suit on Pixiv.*

ure 7. Incorporating real-data-based preference signals on top of FlowGRPO results in more realistic lighting and more natural color distributions, alleviating the stylistic homogenization observed in Figure 2b. These results indicate that

real-data-based preference signals are compatible with supervision derived from human annotations or learned reward models, and can serve as an effective plug-in post-training signal that further improves visual fidelity beyond existing preference alignment pipelines.

*Table 2.* **Constructing preference signals from generated data.** All models are trained starting from the SD-1.5 base model and evaluated on the Pick-a-Pic v2 test set. The proposed two-stage alignment strategy proves effective not only on real data but also when applied to generated data, consistently improving performance over the base model across multiple evaluation metrics. Relative improvements over the SD-1.5 base model are highlighted in (+gain). ImgRwd: ImageReward; UniRwd: UnifiedReward; Aes: LAION aesthetic classifier. Best and second-best results are in **bold** and underlined, respectively.

| # | Method | Preference Score | | | | Image Quality | Aesthetic |
| | | PickScore ↑ | ImgRwd ↑ | UniRwd ↑ | HPSv3 ↑ | DeQA ↑ | Aes ↑ |
|---|---|---|---|---|---|---|---|
| 1 | SD-1.5 | 20.65 | 0.16 | 2.92 | 5.98 | 3.70 | 5.48 |
| 2 | Ours (HPDv3) | 21.04 (+0.39) | 0.38 (+0.22) | 3.06 (+0.14) | 7.33 (+1.35) | 3.96 (+0.26) | 5.64 (+0.16) |
| 3 | Ours (Pick-a-Pic v2) | 21.26 (+0.61) | 0.67 (+0.51) | 3.13 (+0.21) | 7.85 (+1.87) | 3.68 (-0.02) | 5.86 (+0.38) |
| 4 | Ours (Civitai-top) | **21.57** (+0.92) | **0.74** (+0.58) | **3.22** (+0.30) | **8.67** (+2.69) | **4.01** (+0.31) | **5.96** (+0.48) |

*Table 3.* **Ablation study on the two-stage alignment strategy.** The best results are highlighted in **bold**. Pick: PickScore; Aes: LAION aesthetic classifier. Results are reported on the Pick-a-Pic v2 test set with SD-1.5 as the base model.

| Stage 1 | Stage 2 | Pick ↑ | HPSv3 ↑ | DeQA ↑ | Aes ↑ |
|---|---|---|---|---|---|
| ✗ | ✗ | 20.65 | 5.98 | 3.70 | 5.48 |
| ✗ | ✓ | 20.74 | 6.11 | 3.93 | 5.52 |
| ✓ | ✗ | 20.87 | 6.83 | 3.75 | 5.57 |
| ✓ | ✓ | **21.04** | **7.33** | **3.96** | **5.64** |

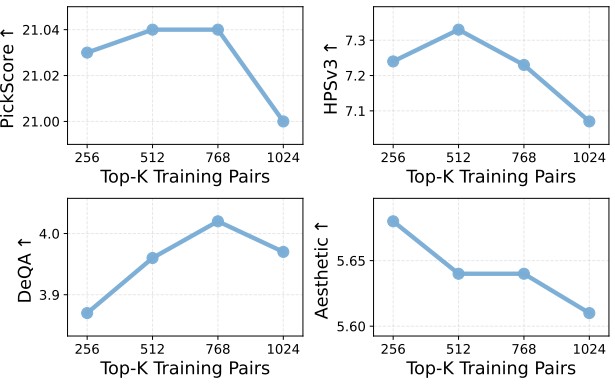

*Figure 8.* **Evaluation results of ours with varying numbers of constructed preference pairs.** Increasing the dataset size from 256 to 512 leads to overall improvement, whereas further increasing the size yields diminishing returns. Results are reported on the Pick-a-Pic v2 test set with SD-1.5 as the base model.

## 4.4. Component Analysis

**Role of the Two-Stage Alignment Strategy.** We next examine the effect of the two-stage alignment strategy by isolating the contribution of each stage, with results summarized in Table 3. When preference optimization is applied directly using the constructed contrastive samples, we observe only limited improvements. A likely reason is that the distribution of real images differs substantially from the model's initial generation distribution, making it difficult for the model to benefit from pairwise supervision alone.

Introducing the distributional warm-up stage substantially improves this behavior. By first encouraging the model to move closer to the distribution represented by real images, the warm-up stage reduces the mismatch between training data and model outputs. This creates a more suitable starting point for subsequent preference learning. Once this alignment is established, applying pairwise optimization becomes more effective, leading to consistent gains across both preference-related and perceptual metrics.

**Effect of Preferred Real Data Size.** We examine how the number of preferred real images affects alignment performance by varying the size of the curated dataset while keeping all other settings fixed. As shown in Figure 8, increasing the number of samples from 256 to 512 leads to consistent improvements across most metrics, indicating that additional high-quality references strengthen the supervision signal. However, further increasing the data size yields diminishing returns. This suggests that alignment

performance is more sensitive to the quality of selected real images than to sheer quantity, and that a relatively small set of well-curated samples is sufficient to provide effective guidance for preference alignment.

**Constructing Preference Signals from Generated Data.** While our main emphasis is on real-image-based supervision, we further investigate whether the proposed two-stage alignment strategy can be applied to preference signals derived from generated data, using the same data curation pipeline. We conduct experiments using preference signals derived from Pick-a-Pic v2 and high-engagement AI-generated images from Civitai-top (WallstoneAI, 2025), where user interactions implicitly reflect preference. Under this setting, we apply the same two-stage alignment procedure without modification. As shown in Table 2, the proposed strategy consistently improves performance over the base model across multiple evaluation metrics.

**Different Perturbation Strategies.** To examine whether the proposed framework depends on a specific negative construction strategy, we evaluate two alternative perturbation strategies, with results summarized in Table 4.

*Table 4.* **Ablation study on the perturbation strategy.** All models are trained starting from the SD-1.5 base model and evaluated on the Pick-a-Pic v2 test set. We study two variants of the perturbation strategy: replacing the default SD-v1.5-Inpainting model with stronger generators (PixArt-$\alpha$ and SD-3.5-M), and constructing negative samples via text-to-image generation conditioned on the real image's caption. Relative improvements over the SD-1.5 base model are highlighted in (+gain). ImgRwd: ImageReward; UniRwd: UnifiedReward; Aes: LAION aesthetic classifier. Best and second-best results are in **bold** and underlined, respectively.

| # | Method | Preference Score | | | | Image Quality | Aesthetic |
| | | PickScore ↑ | ImgRwd ↑ | UniRwd ↑ | HPSv3 ↑ | DeQA ↑ | Aes ↑ |
|---|---|---|---|---|---|---|---|
| 1 | SD-1.5 | 20.65 | 0.16 | 2.92 | 5.98 | 3.70 | 5.48 |
| 2 | Diffusion-DPO | 21.03 (+0.38) | 0.33 (+0.17) | 3.03 (+0.11) | 6.80 (+0.82) | 3.78 (+0.08) | 5.59 (+0.11) |
| 3 | Ours (Text-to-image generation) | **21.05** (+0.40) | 0.35 (+0.19) | 3.04 (+0.12) | 7.09 (+1.11) | **4.00** (+0.30) | **5.65** (+0.17) |
| 4 | Ours (PixArt-$\alpha$ Inpainting) | **21.05** (+0.40) | 0.36 (+0.20) | 3.04 (+0.12) | 7.16 (+1.18) | 3.98 (+0.28) | 5.61 (+0.13) |
| 5 | Ours (SD-3.5-M Inpainting) | 20.96 (+0.31) | 0.35 (+0.19) | 3.04 (+0.12) | 7.20 (+1.22) | 3.89 (+0.19) | 5.62 (+0.14) |
| 6 | Ours (SD-v1.5-Inpainting) | 21.04 (+0.39) | **0.38** (+0.22) | **3.06** (+0.14) | **7.33** (+1.35) | 3.96 (+0.26) | 5.64 (+0.16) |

We first consider stronger inpainting models, replacing SD-v1.5-Inpainting with PixArt-$\alpha$ and SD-3.5-M while keeping the saliency-guided mask selection unchanged. Both variants achieve performance comparable to Diffusion-DPO, suggesting that the method is robust to the choice of inpainting model used for negative construction.

We further replace saliency-guided inpainting entirely with a text-to-image strategy. Specifically, negatives are generated directly from the real image's caption using SD-1.5, without any inpainting or spatial masking. This represents a fundamentally different perturbation mechanism that operates at the global image level rather than the local region level. This alternative still achieves performance comparable to Diffusion-DPO across most evaluation metrics.

Across both local (inpainting-based) and global (generation-based) perturbations, we observe consistent improvements over the base model across all metrics. These results suggest that the effectiveness and robustness of our framework primarily stem from real-image-anchored contrastive learning, rather than a specific negative construction strategy.

## 5. Conclusion

This paper explores preference alignment from a data-centric perspective and investigates whether real images can serve as an effective source of supervision beyond conventional preference pairs. We show that by treating real images as reference points and constructing structured contrasts through controlled perturbations, it is possible to derive informative alignment signals without relying on manually annotated preferences. Across multiple evaluation settings, this approach achieves performance comparable to existing preference-based methods while remaining simple, data-efficient, and complementary to prior alignment strategies. Our findings suggest that real data provide a practical and underutilized source of supervision for preference alignment, offering a promising alternative to relying on large-scale preference annotations or learned reward models.

**Potential limitations and future directions.** While our results demonstrate the effectiveness of real-data-based supervision for preference alignment, several directions remain for future work. Extending the framework to other modalities such as video or multimodal reasoning is a natural next step. In addition, the quality and diversity of reference data play an important role, suggesting the need for more principled data selection strategies. Finally, although our method complements existing preference-based methods, it does not replace human judgment, and integrating real-data supervision with stronger alignment or reward modeling techniques remains an important direction for further improvement.

## Acknowledgements

This work was supported in part by the National Natural Science Foundation of China (NSFC) under Grant Nos. 62376292 and 62536010, the Guangdong Provincial General Fund under Grant No. 2024A1515010208, the Guangzhou Science and Technology Program under Grant Nos. 2025A04J5465 and 2024A04J6365, and the National Road Safety Program under the Australian Government's DITRDCA (Grant No. NRSAGP-TI1-A48). We gratefully acknowledge this support. We also thank all anonymous reviewers for their constructive feedback, which helped improve the quality of this paper.

## Impact Statement

This work explores whether real data itself can serve as a useful source of supervision for preference alignment. The broader societal impacts of such improvements, including positive effects (*e.g.*, enhanced creative applications) and potential negative risks (*e.g.*, misuse of generated images), are consistent with those commonly associated with existing generative image models. The proposed method does not introduce new application scenarios or novel ethical considerations beyond this established context.

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

# A. Implementation Details

### A.1. Hyperparameters Specification

We fine-tune both Stable Diffusion v1.5 (SD-1.5) and Stable Diffusion 3.5 Medium (SD-3.5-M) using LoRA for parameter-efficient adaptation. For SD-1.5, we adopt LoRA with rank $r = 4$ and scaling factor $\alpha = 4$, following prior preference alignment work (Liang et al., 2025). For SD-3.5-M, we use a larger LoRA configuration with $r = 32$ and $\alpha = 64$, consistent with recent alignment method (Liu et al., 2025). All models are trained on four NVIDIA RTX 3090 GPUs. Each GPU processes a batch size per device of one, with gradients accumulated over 64 steps, resulting in an effective batch size of 256.

**Stage 1: Distributional Warm-up.** We employ Diffusion-DRO[1] to steer the models toward the distribution of real images. Most hyperparameters follow the original Diffusion-DRO setup. During training, 20% of text prompts are randomly replaced with empty strings to preserve the model's ability for unconditional generation. The margin parameter $m$ in the Diffusion-DRO objective (Eq. 1) is set to $-0.001$. We train SD-1.5 with a learning rate of $1e^{-4}$ for 1600 optimization steps, while SD-3.5-M with a learning rate of $2e^{-4}$ for 3200 steps. For sampling $x_t$ from the policy model, classifier-free guidance is set to 1.0. SD-1.5 adopts DPMSolver++ with 20 sampling steps, while SD-3.5-M uses FlowMatchEulerDiscreteScheduler with 10 steps.

**Stage 2: Pairwise Preference Learning.** We further align the models using Diffusion-DPO[2] to leverage the fine-grained contrastive signals derived from real data. The learning rate is set to $2.56e^{-6}$ for both models. SD-1.5 employs a constant learning rate schedule with a linear warmup of 125 steps, whereas SD-3.5-M uses a constant schedule without warmup. We set the KL regularization weight $\beta$ to 2000 for SD-1.5 and 100 for SD-3.5-M. The optimization budgets are fixed to 1000 steps for SD-1.5 and 500 steps for SD-3.5-M.

### A.2. Evaluation Settings

To reduce sampling variance and improve the stability of metric estimates, we generate five images per prompt for each model and report the mean score across samples. All training and evaluation are conducted at an image resolution of $512 \times 512$. For image generation during evaluation, we use a classifier-free guidance (CFG) scale of 7.5 with 50 sampling steps for SD-1.5, and a CFG scale of 4.5 with 40 inference steps for SD-3.5-M.

### A.3. User Study Settings

To complement the automated metrics, we conduct a human evaluation study with 18 participants on our fine-tuned SD-3.5-M model. Prompts are sampled from HPDv2 following the protocol of Diffusion-DRO (Wu et al., 2025): 60 prompts spanning four categories (Anime, Concept-Art, Paintings, Photo; 15 each), presented in pairwise randomized order. For each prompt, we conduct two pairwise comparisons: our fine-tuned SD-3.5-M model vs. the original SD-3.5-M model, and our fine-tuned SD-3.5-M model vs. Diffusion-DPO. In each comparison, annotators are presented with two images in randomized order and asked: "Which image do you prefer given the prompt?" We aggregate the annotations at the prompt level. A prompt is counted as a win if the majority of annotators prefer our fine-tuned SD-3.5-M model, as a tie if the votes are evenly split, and as a loss otherwise. The results are reported in Figure 4.

### A.4. Model Specification

Table 5 summarizes the access links for the base models and the various reward models used for evaluation.

### A.5. Saliency-Guided Construction of Contrastive Samples

The construction of contrastive samples follows the pipeline illustrated in Figure 9. First, we apply U$^2$-Net (Qin et al., 2020) to the real image to extract a saliency map. Next, we use the prompt-conditioned SD-v1.5 Inpainting model[3] to regenerate the masked salient regions. Due to the limited generative capability of the inpainting model, the regenerated regions often introduce perceptual artifacts and may not align well with the prompt, resulting in a degraded counterpart. This process establishes a clear contrast between real image and the degraded variant, providing a clear supervision signal for alignment.

---

[1]https://github.com/basiclab/DiffusionDRO
[2]https://github.com/SalesforceAIResearch/DiffusionDPO
[3]https://huggingface.co/stable-diffusion-v1-5/stable-diffusion-inpainting

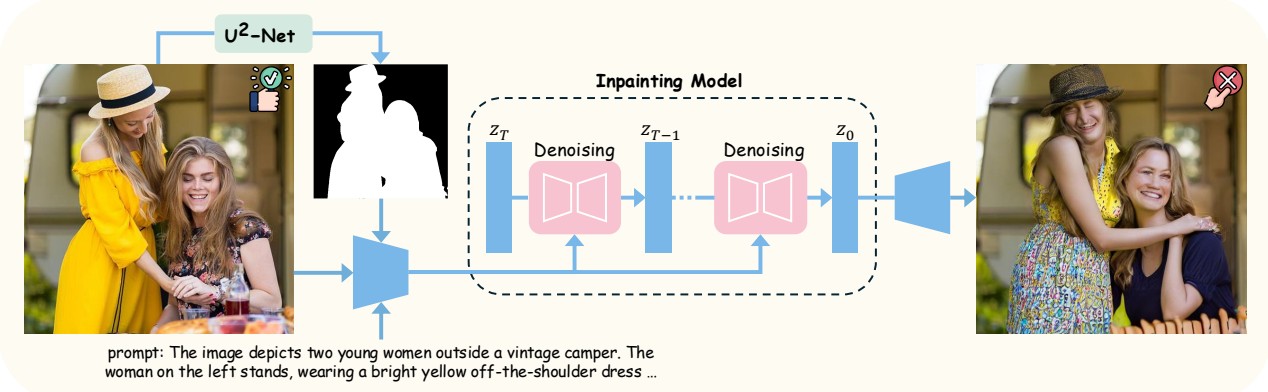

*Figure 9.* **Framework for saliency-guided construction of contrastive samples.** Given a real image, we extract the salient regions using $U^2$-Net (Qin et al., 2020). A prompt-conditioned inpainting model (SD v1.5 Inpainting) regenerates salient regions according to the caption to produce a corresponding degraded counterpart. The resulting discrepancies are localized to salient regions, while the background and global layout are preserved, providing clear contrastive signals for alignment.

*Table 5.* **Model and evaluator resources.** Access links for the base text-to-image models and evaluation models used in our experiments.

| Models | Links |
| --- | --- |
| SD-1.5 | https://huggingface.co/stable-diffusion-v1-5/stable-diffusion-v1-5 |
| SD-3.5-M | https://huggingface.co/stabilityai/stable-diffusion-3.5-medium |
| PickScore | https://huggingface.co/yuvalkirstain/PickScore_v1 |
| ImageReward | https://huggingface.co/THUDM/ImageReward |
| UnifiedReward | https://huggingface.co/CodeGoat24/UnifiedReward-qwen-7b |
| HPSv3 | https://huggingface.co/MizzenAI/HPSv3 |
| DeQA | https://huggingface.co/zhiyuanyou/DeQA-Score-Mix3 |
| LAION aesthetic classifier | https://github.com/LAION-AI/aesthetic-predictor |

In practice, we exclude visually negligible degradations by requiring the PickScore gap between the real and degraded images to exceed 0.02.

## B. Additional Experimental Results and Visualizations

### B.1. Complementarity with Existing Preference Alignment Methods

To verify that the compatibility of real-data-based supervision with existing methods generalizes to diverse prompt distributions, we additionally evaluate this setting on Parti-Prompts (Yu et al., 2022), which is a diverse and compositional prompt benchmark. We apply real-data-based supervision as an additional post-training step on top of models already aligned using Diffusion-DPO or FlowGRPO. As shown in Figure 10, incorporating real-data-based supervision signals consistently improves performance across multiple evaluation metrics. These results are consistent with our observations on the Pick-a-Pic v2 test set and DrawBench, further demonstrating that real-data-based preference signals are compatible with supervision derived from human annotations or learned reward models.

### B.2. Robustness to Curation Choices

To examine whether our method critically depends on data curation, we conduct a systematic ablation that progressively introduces each curation step, with results summarized in Table 6.

Our data curation pipeline applies three steps to real images from HPDv3. First, we use colorfulness filtering to remove visually flat or low-contrast real images, retaining samples with above-average colorfulness scores. Second, for each retained real image, we apply saliency-guided inpainting to generate a degraded counterpart, forming a candidate preference pair. We discard pairs where the degradation is visually negligible, requiring a PickScore gap of at least 0.02 between the real and degraded images to ensure a clear and consistent preference signal. Third, from the remaining candidate pairs, we rank the

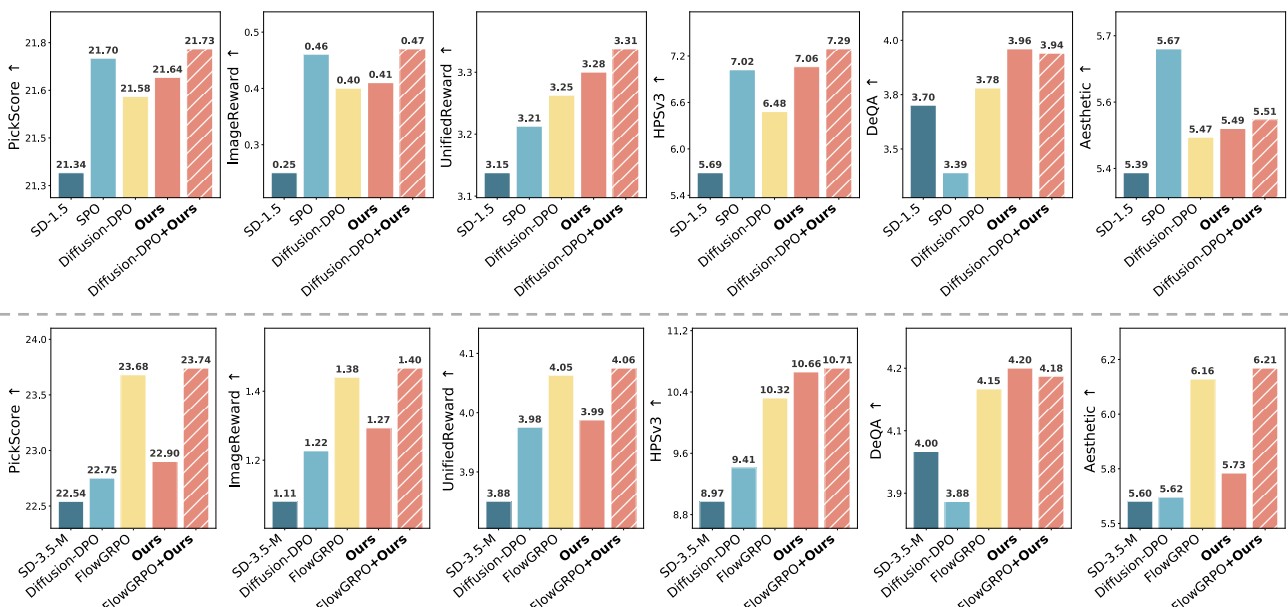

*Figure 10.* **Complementarity with existing preference alignment models.** The evaluation is conducted on Parti-Prompts. Top row: Results based on SD-1.5, where real-data-based supervision is integrated with Diffusion-DPO. Bottom row: Results based on SD-3.5-M, where real-data-based supervision is used as a complementary post-training step on top of FlowGRPO.

*Table 6.* **Ablation study on data curation strategies.** All models are trained starting from the SD-1.5 base model and evaluated on the Pick-a-Pic v2 test set. We ablate three curation components: colorfulness filtering (Color), negligible-degradation filtering (Neg.), and top-512 selection by PickScore (Top-512). Relative improvements over the SD-1.5 base model are highlighted in (+gain). ImgRwd: ImageReward; UniRwd: UnifiedReward; Aes: LAION aesthetic classifier. Best and second-best results are in **bold** and underlined.

| # | Method | Data Curation | | | Preference Score | | | | Image Quality | Aesthetic |
|---|---|---|---|---|---|---|---|---|---|---|
| | | **Color** | **Neg.** | **Top-512** | **PickScore ↑** | **ImgRwd ↑** | **UniRwd ↑** | **HPSv3 ↑** | **DeQA ↑** | **Aes ↑** |
| 1 | SD-1.5 | – | – | – | 20.65 | 0.16 | 2.92 | 5.98 | 3.70 | 5.48 |
| 2 | Diffusion-DPO | – | – | – | 21.03 (+0.38) | 0.33 (+0.17) | 3.03 (+0.11) | 6.80 (+0.82) | 3.78 (+0.08) | 5.59 (+0.11) |
| 3 | No curation | ✗ | ✗ | ✗ | 20.88 (+0.23) | 0.22 (+0.06) | 2.99 (+0.07) | 6.91 (+0.93) | 3.94 (+0.24) | 5.59 (+0.11) |
| 4 | Variant 1 | ✗ | ✗ | ✓ | **21.05** (+0.40) | 0.32 (+0.16) | 3.03 (+0.11) | 7.25 (+1.27) | **3.96** (+0.26) | 5.62 (+0.14) |
| 5 | Variant 2 | ✓ | ✗ | ✓ | 21.02 (+0.37) | 0.35 (+0.19) | 3.04 (+0.12) | 7.19 (+1.21) | 3.95 (+0.25) | **5.64** (+0.16) |
| 6 | Ours | ✓ | ✓ | ✓ | 21.04 (+0.39) | **0.38** (+0.22) | **3.06** (+0.14) | **7.33** (+1.35) | **3.96** (+0.26) | **5.64** (+0.16) |

positive (real) images by PickScore and retain the top-512 pairs as the final training set.

Even without any curation (No curation), the method already improves over the base model and achieves performance comparable to Diffusion-DPO across most metrics. Adding Top-512 selection (Variant 1) yields consistent gains, while further adding colorfulness filtering (Variant 2) and then negligible-degradation filtering (Ours) provides additional but relatively modest improvements.

Overall, the performance improves gradually rather than abruptly as curation is introduced, indicating that the method does not critically depend on highly curated data or specific selection choices. Instead, curation primarily helps stabilize and slightly enhance performance.

### B.3. Preference Signals Learned from Real Images Generalize Beyond Photorealism

The preference signals learned by contrasting real images with generated or perturbed counterparts are not limited to improving realism and photographic fidelity. Professional photographs reflect deliberate human selection and aesthetic judgment, capturing preferences such as composition, semantics, and visual appeal that extend beyond pure photographic fidelity. Our framework thus does not enforce a specific notion of realism, but learns from preference signals encoded in real

*Table 7.* **Preference signals learned from real images generalize beyond photorealism.** (a) User study on non-photorealistic domains. Our user study covers four HPDv2 categories and reports results on Anime and Concept-Art, which are explicitly non-photorealistic. (b) Dense prompt-following ability on DPG-Bench (Hu et al., 2024). Best results are in **bold**.

*(a)* User study on non-photorealistic domains

| Comparison | Anime | Concept-Art |
|---|---|---|
| vs. SD-3.5-M | 73.33 | 73.33 |
| vs. Diffusion-DPO | 60.00 | 66.67 |

*(b)* Dense prompt-following ability on DPG-Bench

| Method | SD-1.5 | SD-3.5-M |
|---|---|---|
| Base | 62.84 | 83.40 |
| Diffusion-DPO | 63.90 | 84.71 |
| Ours | **64.38** | **85.43** |

*Table 8.* **Compute cost of the data curation pipeline.** We report the throughput and total cost of each step measured on a single NVIDIA RTX 3090. The cost is computed under a conservative setting where all 25,096 images are processed before filtering.

| Step | Throughput | Total Cost |
|---|---|---|
| Colorfulness scoring | $\approx$ 4.5 images/s | $\approx$ 1.5 GPU-hours |
| Saliency + Inpainting (50 steps) | $\approx$ 0.14 images/s | $\approx$ 49 GPU-hours |
| PickScore evaluation | $\approx$ 6 pairs/s | $\approx$ 1.2 GPU-hours |
| Total | – | $\approx$ 52 GPU-hours |

data. These signals are general and transferable, as they capture what humans consider visually coherent and appealing.

**User study on non-photorealistic domains.** Our user study covers four HPDv2 categories, including Anime and Concept-Art, which are explicitly non-photorealistic. As shown in Table 7a, our method achieves a 73.33% win rate over the SD-3.5-M base model in both categories, providing direct evidence that real-image-based supervision improves human-perceived quality even in non-photorealistic settings.

**Dense prompt-following ability on DPG-Bench.** We evaluate dense prompt-following ability on DPG-Bench (Hu et al., 2024). Table 7b shows consistent gains over the base model on both SD-1.5 and SD-3.5-M, indicating that real-image-anchored contrastive learning helps the model acquire general perceptual properties such as structural coherence and semantic consistency that transfer beyond photorealistic domains.

**Diversity of model behavior.** Figure 2 shows that our method maintains diversity across attributes such as texture, realism, and stylization, rather than collapsing to a narrow mode. Compared to prior methods that tend to specialize in specific aspects, our model achieves more balanced improvements.

Overall, these results show that our method generalizes preference alignment across diverse visual styles beyond realism.

### B.4. Data and Compute Cost of Constructing the 512 Pairs

We further report the data statistics and compute cost of our curation pipeline.

**Data pipeline.** The pipeline applies three steps to real images from HPDv3, starting from 25,096 samples. First, colorfulness filtering retains samples with above-average colorfulness scores, discarding visually flat or low-contrast images, yielding 10,475 images. Second, for each retained image, we apply saliency-guided inpainting ($U^2$-Net + SD-v1.5-Inpainting) to generate a degraded counterpart and form a candidate preference pair. We discard pairs with visually negligible degradations (PickScore gap below 0.02) to ensure a clear and consistent preference signal, resulting in 4,041 candidate pairs. Third, we rank the remaining pairs by the PickScore of the real images and retain the top-512 pairs as the final training set. All steps are fully automatic and require no human annotation.

**Compute cost.** We measure the cost of each step on a single NVIDIA RTX 3090, conservatively reporting the cost of processing the full 25,096 images before any filtering. As summarized in Table 8, the entire curation pipeline takes only about 52 GPU-hours, indicating that real-data-based preference signals can be obtained at low cost.

### B.5. Additional Qualitative Comparisons

Figures 11 and 12 present further qualitative results. These visualizations demonstrate that real images serve as an effective source of supervision for preference alignment, eliminating the need for manually annotated preference pairs. Furthermore,

| SD-1.5 | SPO | Diffusion-DPO | Ours | Diffusion-DPO+Ours |
|---|---|---|---|---|

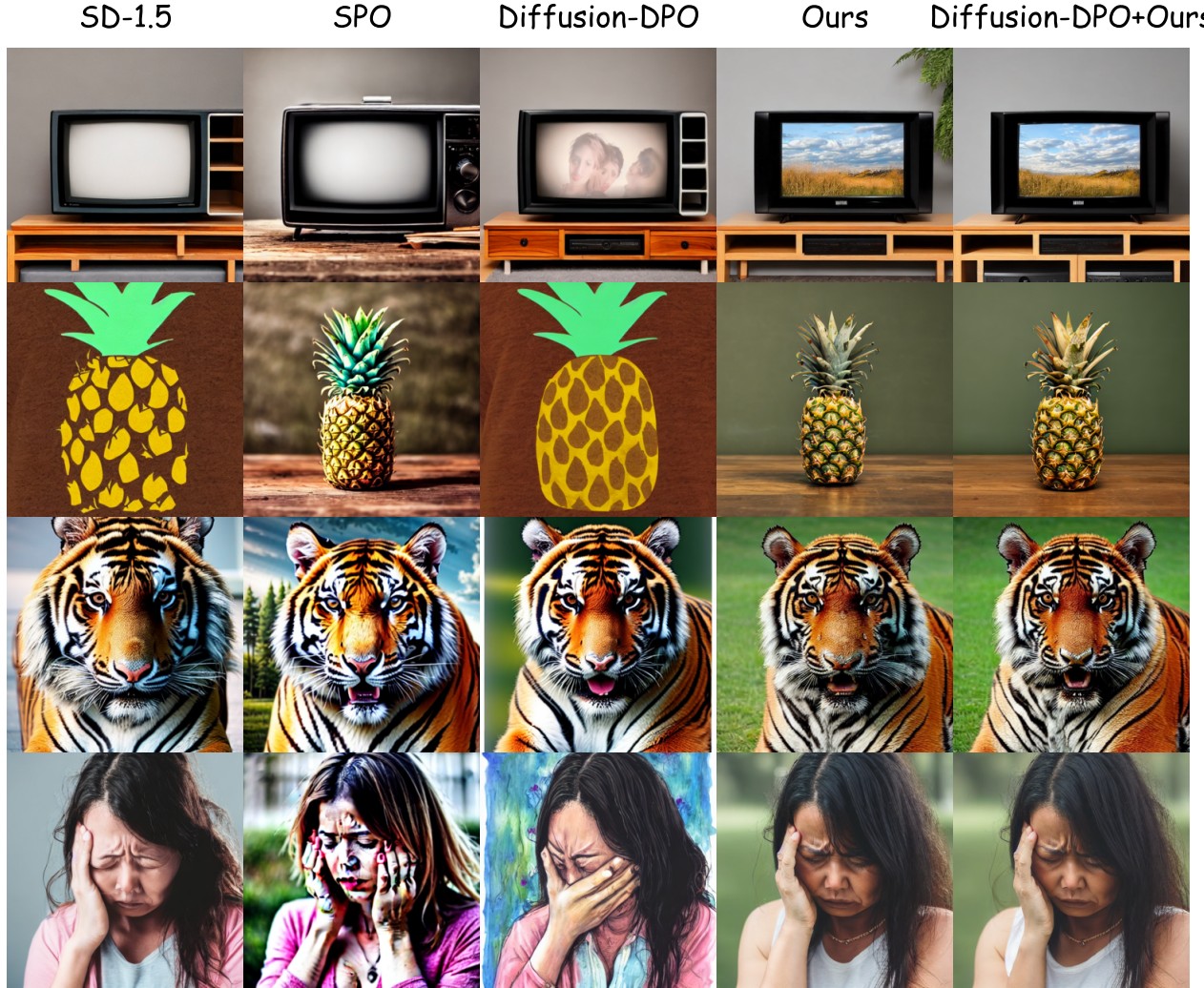

*Figure 11.* **Qualitative comparison based on SD-1.5.** Post-training with real data produces images with improved visual realism and richer texture details. When applied as an additional post-training stage on top of Diffusion-DPO (Wallace et al., 2024), it also improves the visual realism of the resulting generations. Prompts from top to bottom : (1) *a TV.* (2) *a pineapple.* (3) *a tiger cow.* (4) *30 year old short slim man, fuller round face, very short hair, black hair, black stubble, olive skin, immense detail/ hyper. Pårealistic, city /cyberpunk, high detail, detailed, 3d, trending on artstation, cinematic.*

our results show that real-data-based supervision achieves competitive performance and serves as a robust complementary post-training step to existing preference-based alignment methods. This integration improves the realism of generation results and yields more natural color distributions.

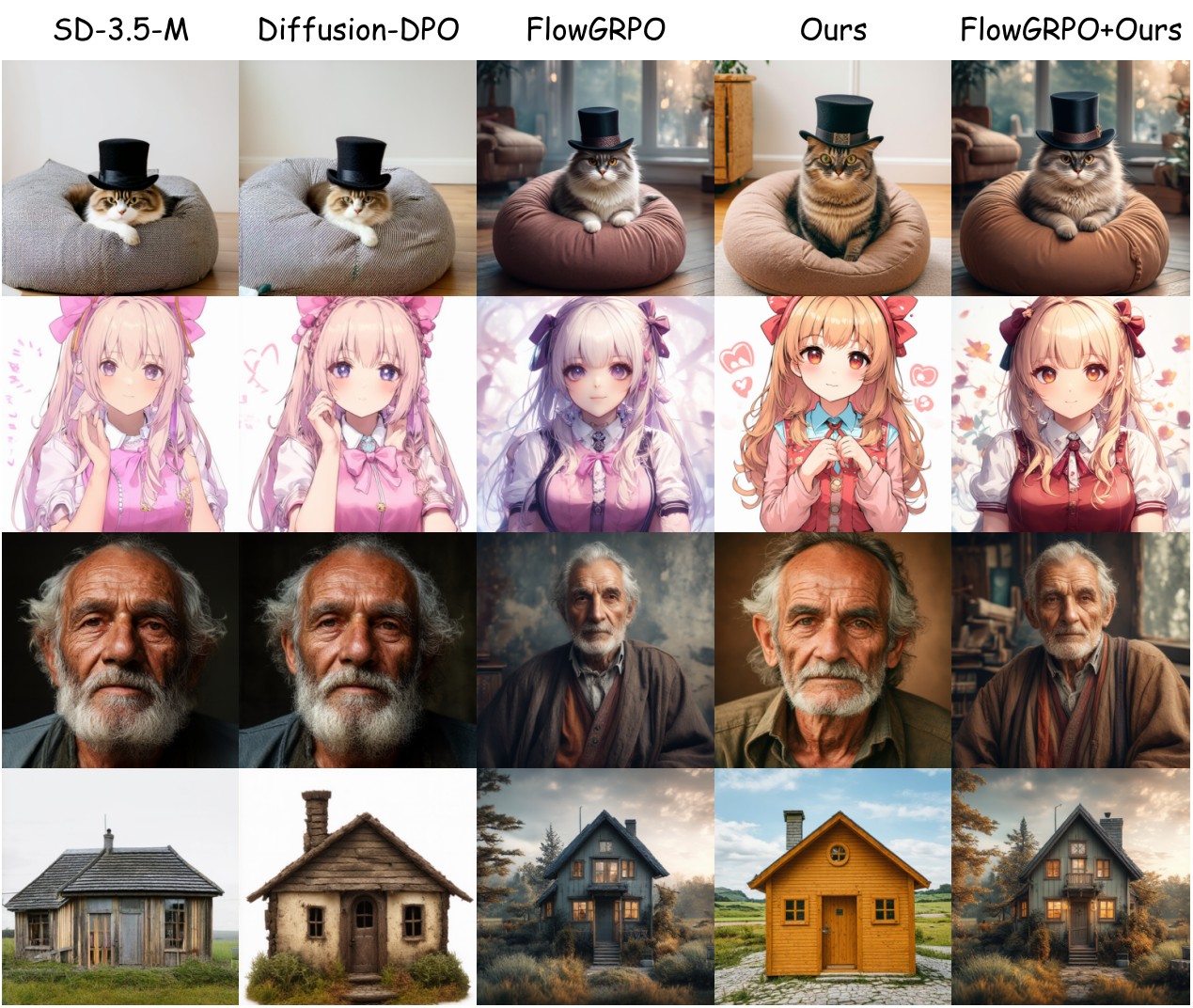

*Figure 12.* **Qualitative comparison based on SD-3.5-M.** Compared to FlowGRPO (Liu et al., 2025), post-training with real data yields more realistic lighting and more natural color distributions. When applied as an additional post-training step on top of FlowGRPO, it further alleviates stylistic homogenization and enhances texture details. Prompts from top to bottom : (1) *Cat with a top hat on a bean bag.* (2) *cute waifu.* (3) *a portrait of an old man.* (4) *a small house.*

