# OpenReview forum: "When Preference Labels Fall Short: Aligning Diffusion Models from Real Data"
_ICML.cc/2026/Conference — ICML 2026 regular_

### Official Review · Reviewer_Q2fi · 2026-03-11

**Soundness:** 3
**Presentation:** 3
**Significance:** 3
**Originality:** 3
**Overall Recommendation:** 4
**Confidence:** 4

**Summary:**

This paper studies whether diffusion-model preference alignment can be performed without manually annotated preference pairs. The core idea is to use real images as implicit positive references and construct preference supervision by contrasting them with either model-generated samples or controlled degradations of those real images.

Experiments on SD-1.5 and SD-3.5-M show that this real-data-based supervision can achieve performance comparable to preference-pair-based alignment

**Compliance With Llm Reviewing Policy:**

Affirmed.

**Final Justification:**

The authors substantively addressed several of my main concerns.

**Key Questions For Authors:**

1. How well does the method generalize beyond realism-oriented alignment?

2. What is the actual data and compute cost of constructing the 512 pairs?

3. How much does the method depend on the choice of curated real images?

**Limitations:**

Yes

**Strengths And Weaknesses:**

Strengths:

1. The paper addresses a relevant and timely question in diffusion-model alignment: whether high-quality real data can substitute for manually annotated preference pairs, which are expensive and often imperfect.

2. The main contribution is conceptually simple and easy to understand. Using real images as anchors, then generating degraded contrasts via saliency-guided inpainting, is an intuitive data-centric alternative to relying entirely on pairwise human labels.

Weaknesses:

1. The paper argues that real-data-based supervision is an effective alternative to preference-label-based supervision, but most evidence is based on automated evaluation metrics such as PickScore, ImageReward, UnifiedReward, HPSv3, and DeQA. These metrics are themselves imperfect proxies, and some are closely related to the same preference datasets or reward-model paradigms that the paper critiques. As a result, it remains unclear whether the proposed method is truly aligning better with human preference, or simply optimizing toward existing learned evaluators.

2. Diffusion-DPO is compared using 851k labeled pairs, while the proposed method uses 512 curated pairs, which is impressive on paper. However, the paper’s 512 pairs are selected from professional photographs in HPDv3 and further filtered by colorfulness and PickScore gap criteria, so the supervision quality is much higher and more curated. This makes the result interesting, but it also means the comparison is not purely about annotation efficiency.

3. The positive references are drawn from professional photographs, and the degradation pipeline is based on saliency-guided inpainting. This seems naturally suited to improving realism and photographic fidelity, but it is less clear whether it generalizes to broader alignment goals such as stylization, compositional faithfulness, creative preference, or non-photographic domains.

4. Performance improvement is marginal. As shown in Tables 1 and 2, the proposed method offers little improvement compared to Diffusion-DPO.

---

> ### Author Rebuttal · Authors · 2026-03-31
>
> > **Q1: Human evaluation**
>
> Our method achieved a win rate of 63.33% over the SD-3.5-M baseline and 56.67% over Diffusion-DPO, consistent with automated metric findings. Please refer to our response to Reviewer K2hq (Q3) for full details.
>
> > **Q2-1: Annotation efficiency and data quality**
>
> Our curated pairs are intentionally higher-quality. Our goal is not annotation-efficiency comparison, but to show that strong preference signals can be constructed automatically without human labeling. Our fully automatic pipeline achieves competitive performance with only 512 pairs, versus 851k human-labeled pairs for Diffusion-DPO.
>
> > **Q2-2: Data and compute cost**
>
> **Data pipeline.** Starting from 25,096 real images in HPDv3, we retain 10,475 images after colorfulness filtering, and obtain 4,041 candidate pairs after saliency-guided inpainting and discarding visually negligible degradations. From these, we select the final 512 training pairs by PickScore ranking.
>
> **Compute cost.** All steps are measured on a single NVIDIA RTX 3090 (processing all 25,096 images before filtering):
> |Step|Throughput|Total Cost|
> |---|:----|:----|
> |Colorfulness scoring|≈ 4.5 image/s|≈ 1.5 GPU-hours|
> |Saliency + Inpainting (50 steps)|≈ 0.14 image/s|≈ 49 GPU-hours|
> |PickScore evaluation|≈ 6 pairs/s|≈ 1.2 GPU-hours|
> |Total||≈ 52 GPU-hours|
>
> > **Q3: Generalization beyond realism**
>
> We clarify that professional photographs encode broader human preferences, such as composition, semantics, and visual appeal, not just realism. Our framework does not enforce a specific notion of realism, but learns from preference signals encoded in real data. These signals are general and transferable, as they capture what humans consider visually coherent and appealing, as evidenced below.
>
> **User study on non-photorealistic domains.** Our user study covers four HPDv2 categories, including Anime and Concept-Art, which are non-photorealistic. Our method achieves win rates of 73.33% over the SD-3.5-M base model in both categories.
>
> |Win rates (%)|Anime|Concept-Art|
> |---|:---:|:---:|
> |Vs. SD-3.5-M|73.33|73.33|
> |Vs. Diffusion-DPO|60.00|66.67|
>
> **Prompt-following ability on DPG-Bench.** We evaluate on DPG-Bench, which measures semantic alignment between prompts and generated images. Our method consistently improves over the base model on both SD-1.5 and SD-3.5-M.
>
> |Method|SD-1.5|SD-3.5-M|
> |---|:---:|:---:|
> |Base|62.84|83.40|
> |Diffusion-DPO|63.90|84.71|
> |Ours|**64.38**|**85.43**|
>
> **Diversity of model behavior.** Fig. 2 shows that our method maintains diversity across attributes such as texture, realism, and stylization, rather than collapsing to a narrow mode.
>
> Overall, these results show that real-image-based supervision generalizes beyond photorealistic domains.
>
> > **Q4: Dependency on curated images**
>
> We present a systematic ablation to test sensitivity to each curation step: colorfulness filtering to exclude low-contrast samples (Line 192), negligible-degradation filtering via PickScore gap < 0.02 to ensure informative pairs (Lines 218, 650), and top-512 selection by PickScore (Line 229).
>
> **Curation ablation**
> |Method|Color.Filter|Neg.Discard|Top-512|PickScore|ImgRwd|UniRwd|HPSv3|DeQA|Aes.|
> |---|:---:|:---:|:---:|:---:|:---:|:---:|:---:|:---:|:---:|
> |SD-1.5|-|-|-|20.65|0.16|2.92|5.98|3.70|5.48|
> |Diffusion-DPO|-|-|-|21.03|0.33|3.03|6.80|3.78|5.59|
> |No curation|✗|✗|✗|20.88|0.22|2.99|6.91|3.94|5.59|
> |Variant 1|✗|✗|✓|21.05|0.32|3.03|7.25|3.96|5.62|
> |Variant 2|✓|✗|✓|21.02|0.35|3.04|7.19|3.95|5.64|
> |Ours|✓|✓|✓|21.04|0.38|3.06|7.33|3.96|5.64|
>
> Even without any curation (“No curation”), the method already improves over the base model and achieves performance comparable to Diffusion-DPO across most metrics. Adding Top-512 selection (Variant 1) yields consistent gains, while colorfulness filtering and negligible-degradation filtering (Variant 2) provide additional but relatively modest improvements.
>
> Overall, performance improves gradually as curation is introduced, indicating the method does not critically depend on highly curated data. Curation primarily helps stabilize and slightly enhance performance.
>
> > **Q5: Marginal improvement over Diffusion-DPO**
>
> We clarify the performance improvement from four perspectives.
>
> First, our method achieves larger improvements over the base model than Diffusion-DPO, indicating stronger alignment gains under the same backbone.
>
> Second, per-sample win rates confirm clear advantages over Diffusion-DPO, consistent with human evaluation (Q1).
> |Win rates (%)|PickScore|ImgRwd|HPSv3|DeQA|Aes.|
> |---|:---:|:---:|:---:|:---:|:---:|
> |Vs. Diffusion-DPO (SD-1.5)|54.31|50.18|57.64|70.60|51.26|
> |Vs. Diffusion-DPO (SD-3.5-M)|60.75|60.64|74.60|84.08|63.96|
>
> Third, qualitative improvements are perceptually significant, with better realism, richer textures, and more natural colors (Figs. 5, 6, 10, 11).
>
> Finally, our method is complementary: stacking it on top of Diffusion-DPO (Fig. 4) yields further gains, capturing signals beyond labeled pairs.

---

> > ### Author Rebuttal · Reviewer_Q2fi · 2026-04-03
> >
> > Thank you for the rebuttal. The authors substantively addressed several of my main concerns by adding human evaluation, clarifying the data construction and compute cost, and providing further evidence on generalization beyond photorealistic settings and sensitivity to curation choices. While I still have some reservations about the scope of the human study and the exact fairness of the comparison to Diffusion-DPO, the rebuttal provides meaningful new evidence that improves my assessment of the paper. Therefore, I have decided to raise my score from Weak Reject to Weak Accept.

---

> > > ### Author Response · Authors · 2026-04-03
> > >
> > > Dear Reviewer Q2fi,
> > >
> > > Thank you for the thoughtful follow-up and for revising the score. We are glad that the additional evidence helped clarify our contributions.
> > >
> > > We appreciate the reviewer’s suggestions. In the revision, we will incorporate the clarifications and experiments from the rebuttal.
> > >
> > > We will clarify that the comparison shows curated data can encode preference signals without human annotation while achieving competitive performance, rather than focusing solely on annotation efficiency. It indicates a practical trade-off: competitive performance can be achieved with a small number of automatically curated samples instead of large-scale labeled pairs, using simple criteria (e.g., colorfulness, PickScore gap) without too much human intervention.
> > >
> > > Human evaluation results will be included, following the protocol of Diffusion-DRO, across both photorealistic and non-photorealistic categories.
> > >
> > > Best,
> > >
> > > The Authors

---

### Official Review · Reviewer_4Fzq · 2026-03-12

**Soundness:** 3
**Presentation:** 4
**Significance:** 3
**Originality:** 3
**Overall Recommendation:** 4
**Confidence:** 4

**Summary:**

This paper proposes a data-centric approach to diffusion-model alignment. Instead of relying only on preference pairs constructed from generated images, the method uses curated real images as anchors, creates degraded counterparts via saliency-guided inpainting, and trains with a two-stage pipeline consisting of distribution alignment followed by contrastive preference learning. With only 512 constructed pairs, the paper reports performance comparable to or better than Diffusion-DPO trained on 851k preference pairs. For example, on SD-1.5 it reports PickScore 21.04 versus 21.03 and HPSv3 7.33 versus 6.80. The paper also shows that the method can be used as a complementary post-training step on top of existing pipelines such as Diffusion-DPO and FlowGRPO.

**Compliance With Llm Reviewing Policy:**

Affirmed.

**Key Questions For Authors:**

1. Can the authors provide any direct human evaluation, even on a small subset, to verify that the metric gains correspond to human preferences? A convincing answer here would improve my soundness assessment.
2. How sensitive are the results to the specific saliency-guided inpainting process used to construct negatives? Evidence that the method is robust to alternative perturbation strategies would increase my confidence in the generality of the approach.
3. Can the authors provide a more tightly matched generated-data baseline under a similar data budget and training schedule? If the proposed method still holds up under that comparison, it would strengthen my originality and significance assessments.
4. How robust is the method when the reference image set is less curated or contains lower-quality images? A strong answer here would improve my view of the practical significance of the method.

**Limitations:**

Yes.

The paper includes a reasonable limitations discussion and explicitly notes both the importance of reference-data quality and the fact that the method does not replace human judgment.

**Strengths And Weaknesses:**

Strengths

1. Clear and well-motivated central idea

The paper makes a clean argument that supervision quality, not only objective design, can be a bottleneck in diffusion-model preference alignment. Using real images as reference anchors is a natural and practically meaningful alternative to relying entirely on generated preference pairs.

2. Strong data efficiency

The empirical message is compelling: the method remains competitive while using only 512 constructed pairs, whereas Diffusion-DPO is trained with 851k preference pairs. Even allowing for imperfect comparability, this is a notable result.

3. Reasonable two-stage design with supporting ablation

The distribution-alignment warm-up is well motivated by the gap between real images and model outputs, and the ablation table supports the claim that both stages contribute to final performance.

4. Practical complementarity

The method is not presented only as a replacement for prior approaches. The post-training results on top of Diffusion-DPO and FlowGRPO make the paper more practically relevant, since the approach can be layered onto existing alignment pipelines.

5. Good presentation of the main empirical story

The paper is generally easy to follow. The method description, the two-stage ablation, and the complementary post-training experiments are organized clearly enough that the central claim is easy to track.

Weaknesses

1. Lack of direct human evaluation

For a preference-alignment paper, the evaluation relies entirely on automated metrics such as PickScore, HPSv3, ImageReward, and BERT-based measures. This is the biggest empirical gap. Metric improvements are encouraging, but they are not a full substitute for human preference judgments.

2. Specific negative-sample construction

The contrastive supervision is built through saliency-guided inpainting. This is plausible, but it is still a fairly specific corruption process and may capture only a subset of the failure modes that matter for real human preferences.

3. Comparison with Diffusion-DPO is informative but not tightly matched

The headline comparison is interesting, but the training data source and scale differ substantially. As a result, it remains somewhat unclear how much of the gain comes from the proposed data construction itself versus broader differences in supervision setup.

4. Limited robustness analysis for reference-data quality

The paper studies the number of preferred real images and shows diminishing returns beyond 512 samples, but it says less about robustness when the reference set is noisier or less curated. This matters because the paper itself notes that the quality and diversity of the reference data play an important role.

5. Failure-case analysis is limited

The paper would be stronger with a more explicit discussion of when the method underperforms or where the real-data supervision signal is least helpful.

---

> ### Author Rebuttal · Authors · 2026-03-31
>
> > **Q1: Human evaluation**
>
> Our method achieved a win rate of 63.33% over the SD-3.5-M baseline and 56.67% over Diffusion-DPO, consistent with automated metric findings. Please refer to our response to Reviewer K2hq (Q3) for full details.
>
> > **Q2: Sensitivity to inpainting strategy**
>
> Our method is not tied to a specific saliency-guided inpainting strategy; rather, this process serves as one practical way to construct informative negative samples. To assess sensitivity, we further evaluate two alternative perturbation strategies:
>
> - Stronger inpainting models. We replace SD-v1.5-Inpainting with PixArt-α and SD-3.5-M while keeping the saliency-guided mask selection unchanged. Both achieve performance comparable to Diffusion-DPO, suggesting the method is robust to the choice of inpainting model.
>
> - Text-to-image generation. We replace saliency-guided inpainting entirely with a text-to-image strategy. Negatives are generated from the real image's caption using SD-1.5, without any inpainting or spatial masking. This represents a fundamentally different perturbation mechanism that operates at a global level. As shown in the table below, this alternative still achieves performance comparable to Diffusion-DPO across most metrics.
> |Model|PickScore|ImgRwd|UniRwd|HPSv3|DeQA|Aes.|
> |---|:---:|:---:|:---:|:---:|:---:|:---:|
> |SD-1.5|20.65|0.16|2.92|5.98|3.70|5.48|
> |Diffusion-DPO|21.03|0.33|3.03|6.80|3.78|5.59|
> |Ours (Text-to-image generation)|21.05|0.35|3.04|7.09|4.00|5.65|
> |Ours (SD-v1.5-Inpainting)|21.04|0.38|3.06|7.33|3.96|5.64|
> |Ours (PixArt-α Inpainting)|21.05|0.36|3.04|7.16|3.98|5.61|
> |Ours (SD-3.5-M Inpainting)|20.96|0.35|3.04|7.20|3.89|5.62|
>
> Consistent improvements across both local and global perturbation strategies confirm that the gains stem from real-image-anchored contrastive learning, not a specific negative construction method.
>
>
> > **Q3: Matched generated-data baseline**
>
> We thank the reviewer for this constructive suggestion. Table 2 directly addresses this request. Row 3 (Ours, Pick-a-Pic v2) uses generated images from Pick-a-Pic v2 as reference anchors without relying on any human preference labels, processed through the same data curation pipeline. This is a tightly controlled comparison: Pick-a-Pic v2 is the same dataset used to train Diffusion-DPO.
>
> As shown in Table 2, Ours (Pick-a-Pic v2, 512 pairs) achieves competitive or better performance compared to Diffusion-DPO (851k pairs) across most metrics (e.g., PickScore 21.26 vs. 21.03 and Aesthetic Score 5.86 vs. 5.59).
>
> > **Q4: Robustness to reference image quality**
>
> We clarify the data curation pipeline, then present a systematic ablation to directly test sensitivity to each curation step.
>
> **Data curation pipeline.**
>
> 1. **Colorfulness filtering**: We apply a lightweight colorfulness-based filter and retain images with above-average scores as a heuristic to remove visually flat or low-contrast samples (Line 192).
> 2. **Negligible-degradation filtering**: For each real image, we apply saliency-guided inpainting to construct a degraded counterpart, forming a candidate preference pair. We discard pairs where the degradation is visually negligible, requiring a PickScore gap of at least 0.02 between the real and degraded images to ensure a clear and consistent preference signal (Lines 218, 650).
> 3. **Top-512 selection by PickScore**: From the remaining candidate pairs, we rank positive (real) images by PickScore and retain the top 512 (Line 229).
>
> **Curation ablation.**
> |Method|Color.Filter|Neg.Discard|Top-512|PickScore|ImgRwd|UniRwd|HPSv3|DeQA|Aes.|
> |---|:---:|:---:|:---:|:---:|:---:|:---:|:---:|:---:|:---:|
> |SD-1.5|-|-|-|20.65|0.16|2.92|5.98|3.70|5.48|
> |Diffusion-DPO|-|-|-|21.03|0.33|3.03|6.80|3.78|5.59|
> |No curation|✗|✗|✗|20.88|0.22|2.99|6.91|3.94|5.59|
> |Variant 1|✗|✗|✓|21.05|0.32|3.03|7.25|3.96|5.62|
> |Variant 2|✓|✗|✓|21.02|0.35|3.04|7.19|3.95|5.64|
> |Ours|✓|✓|✓|21.04|0.38|3.06|7.33|3.96|5.64|
>
> Even without any curation (“No curation”), the method already improves over the base model and achieves performance comparable to Diffusion-DPO across most metrics. Adding Top-512 selection (Variant 1) yields consistent gains, while colorfulness filtering and negligible-degradation filtering (Variant 2) provide additional but relatively modest improvements.
>
> Overall, the performance improves gradually rather than abruptly as curation is introduced, indicating that the method does not critically depend on highly curated data. Instead, curation primarily helps stabilize and slightly enhance performance.
>
> > **Q5: Failure cases analysis**
>
> Good suggestion. Our reference images are primarily natural photographs, which rarely contain text. As a result, the supervision signal provides limited guidance for text-related attributes, and we observe that our method does not significantly improve the model’s ability to generate legible or semantically accurate text. Addressing this limitation would likely require incorporating text-rich or OCR-aware supervision.

---

### Official Review · Reviewer_X4Rx · 2026-03-13

**Soundness:** 2
**Presentation:** 2
**Significance:** 2
**Originality:** 1
**Overall Recommendation:** 3
**Confidence:** 4

**Summary:**

This work studies whether real images can be used as supervision for preference alignment in generative models, instead of relying only on paired comparisons between generated samples. Existing preference-based methods often depend on relative judgments between two imperfect generated images, which can make the supervision noisy and ambiguous. To address this, the paper proposes a data-centric curation strategy that uses real images as anchors and builds preference signals by contrasting them with generated or perturbed samples, without manual preference annotations. Experiments show that this real-data-based supervision can guide diffusion models effectively and reach performance comparable to standard preference alignment methods. Overall, the results suggest that real data is a practical and complementary source of supervision for more label-efficient alignment.

**Compliance With Llm Reviewing Policy:**

Affirmed.

**Key Questions For Authors:**

1. **Why are different test sets used for different generative models?**
   For example, why is Pick-a-Pic v2 used for SD v1.5, while DrawBench is used for SD v3.5? It would be helpful to clarify the rationale for this choice and discuss whether the difference in evaluation sets affects the fairness of the comparison.

2. **How were the 512 training pairs in Table 1 selected?**
   Are these pairs a subset of the pairs contained in HPDv3? If so, what criteria were used for selection? It would also be useful to know how sensitive the results are to this choice. For instance, whether similar performance is obtained with different randomly sampled or curated sets of 512 pairs.

3. **Could you clarify the experimental setup for Table 2?**
   If generated data are used in this experiment, it is not entirely clear how this setting differs from existing preference alignment approaches. A more explicit explanation of the data construction and the distinction from prior work would make the contribution easier to understand.

**Limitations:**

yes

**Strengths And Weaknesses:**

### Strengths
1. The background, preliminaries, and method are presented clearly.
The paper is generally well organized, and the technical components are introduced in a way that makes the overall approach easy to follow.

### Weaknesses
1. The method appears to equate alignment with realism, which may be an overly narrow assumption.

2. The proposed approach encourages the generative model to produce images that are closer to real photographs. However, realism is not always the preferred objective in image generation, since users may value other attributes such as stylization, creativity, or abstraction. In that sense, the paper seems to assume that realism is universally desirable, and I do not find this assumption fully convincing.
The experimental setup is not sufficiently clear.
In particular, the statement “We select the top 512 preference pairs based on the PickScore (Kirstain et al., 2023) of the positive samples” is difficult to interpret in the context of the overall pipeline. It is not clear at which stage these 512 preference pairs are used or how they relate to the main training data. From the current description, I only understand that real images from HPDv3 and their artificially degraded counterparts are used, so the role of these selected pairs should be explained more explicitly.

3. The discussion of data size and curation raises questions about robustness and hyperparameter sensitivity.
The paper states that increasing the amount of real data leads to diminishing returns, and argues that alignment performance depends more on the quality of selected real images than on the quantity. However, this claim would benefit from deeper analysis. As presented, the method appears to rely heavily on the selection and number of real images, which raises the concern that the results in Table 1 may depend strongly on tuned curation choices or favorable hyperparameter settings. More systematic analysis would help clarify whether the reported gains are robust.

---

> ### Author Rebuttal · Authors · 2026-03-31
>
> > **Q1: Realism as an overly narrow alignment objective**
>
> Our work does **not** assume realism is universally desirable. Professional photographs encode broader human preferences (composition, semantics, and visual appeal) through deliberate aesthetic selection, not merely realism. Our framework does not enforce a specific notion of realism, but learns from preference signals encoded in real data. These signals are general and transferable, as they capture what humans consider visually coherent and appealing, as evidenced below.
>
> **User study on non-photorealistic domains.** Our method achieves win rates of 73.33% over the SD-3.5-M base model in both Anime and Concept-Art categories. Please see our response to Reviewer K2hq (Q3) for details.
> |Win rates (%)|Anime|Concept-Art|
> |---|:---:|:---:|
> |Vs. SD-3.5-M|73.33|73.33|
> |Vs. Diffusion-DPO|60.00|66.67|
>
> **Prompt-following ability on DPG-Bench.** DPG-Bench measures semantic alignment between text and generated images; our method consistently improves over the base model on both SD-1.5 and SD-3.5-M.
> |Method|SD-1.5|SD-3.5-M|
> |---|:---:|:---:|
> |Base|62.84|83.40|
> |Diffusion-DPO|63.90|84.71|
> |Ours|**64.38**|**85.43**|
>
> **Diversity of model behavior.** Fig. 2 shows that our method maintains diversity across attributes like texture, realism, and stylization, rather than collapsing to a narrow mode.
>
> > **Q2-1: Selection of the 512 training pairs**
>
> **The 512 pairs are not human-annotated preference pairs from HPDv3, but constructed via our data curation pipeline from real images sampled from HPDv3:**
>
> 1. We apply a lightweight colorfulness-based filter to remove visually flat / low-contrast samples (Line 192).
> 2. For each retained real image, we apply saliency-guided inpainting to generate a degraded counterpart, forming a candidate preference pair. We discard pairs where the degradation is visually negligible, requiring a PickScore gap of at least 0.02 between the real and degraded images to ensure a clear and consistent preference signal (Lines 218, 650).
> 3. From this pool, we rank pairs by PickScore of the real image and keep the top 512 pairs (Line 229).
>
> The 512 pairs serve as the entire training set: the positive (real) images anchor Stage 1 (distribution alignment), and the full pairs are used in Stage 2 (contrastive preference optimization).
>
> > **Q2-2: Sensitivity to curation choices**
>
> We present a systematic ablation to test sensitivity to each curation step: colorfulness filtering, negligible-degradation filtering (PickScore gap < 0.02), and top-512 selection by PickScore.
>
> **Curation ablation**
> |Method|Color.Filter|Neg.Discard|Top-512|PickScore|ImgRwd|UniRwd|HPSv3|DeQA|Aes.|
> |---|:---:|:---:|:---:|:---:|:---:|:---:|:---:|:---:|:---:|
> |SD-1.5|-|-|-|20.65|0.16|2.92|5.98|3.70|5.48|
> |Diffusion-DPO|-|-|-|21.03|0.33|3.03|6.80|3.78|5.59|
> |No curation|✗|✗|✗|20.88|0.22|2.99|6.91|3.94|5.59|
> |Variant 1|✗|✗|✓|21.05|0.32|3.03|7.25|3.96|5.62|
> |Variant 2|✓|✗|✓|21.02|0.35|3.04|7.19|3.95|5.64|
> |Ours|✓|✓|✓|21.04|0.38|3.06|7.33|3.96|5.64|
>
> Even without any curation (“No curation”), the method already improves over the base model and matches Diffusion-DPO across most metrics. Adding Top-512 selection (Variant 1) yields consistent gains, while colorfulness filtering and negligible-degradation filtering (Variant 2) provide additional but relatively modest improvements.
>
> Figure 7 evaluates different data budgets: performance improves steadily with more data and then saturates, showing diminishing returns rather than instability. Across different data sizes, our method consistently outperforms the base model.
>
> Overall, the gradual improvement indicates the method does not critically depend on highly curated data. Curation primarily stabilizes and slightly enhances performance.
>
> > **Q3: Different test sets for different models**
>
> We follow established practice: SD-1.5 uses Pick-a-Pic v2 (following Diffusion-DRO); SD-3.5-M uses DrawBench (following FlowGRPO), ensuring direct comparability with the most relevant baselines.
>
> We further evaluate both models on a shared benchmark (PartiPrompts, Table 1(b)) and observe consistent improvements under this unified setting.
>
> > **Q4: Experimental setup for Table 2**
>
> Table 2 evaluates whether our framework remains effective when reference anchors are high-quality generated images rather than real photographs (Line 418).
>
> - Data construction: we use high-quality generated images from public datasets (e.g., Civitai-top) as reference anchors, applying the same curation pipeline. No human preference annotations are used.
> - Distinction from prior work: Unlike prior methods (e.g., Diffusion-DPO) that rely on pairwise human judgments, our framework constructs preference signals via controlled perturbation: pairing a reference image with its inpainted degraded counterpart, without any human annotations.
>
> Table 2 thus shows that our effectiveness stems from the contrastive construction mechanism, not from the specific origin of the reference images.

---

### Official Review · Reviewer_K2hq · 2026-03-13

**Soundness:** 2
**Presentation:** 3
**Significance:** 2
**Originality:** 2
**Overall Recommendation:** 4
**Confidence:** 3

**Summary:**

The paper introduces a two-stage framework for preference optimization of diffusion models. It first anchors the model to a real-world photographic distribution using Distribution-DRO and then refines local details using contrastive pairs created via saliency-guided inpainting. The method is shown to have much better sample efficiency compared with only using synthetic preference pairs.

**Compliance With Llm Reviewing Policy:**

Affirmed.

**Final Justification:**

As mentioned in Rebuttal Acknowledgement, the results from different inpainting models and human eval addressed my concerns. But I am not sure whether the method could be a generalized approach for preference learning, thus I change my score to weak accept.

**Key Questions For Authors:**

1. Is there human evaluation results instead of the metrics trained on the synthetic images like PickScore and HPS?
2. Using only 512 samples to ground the model sounds like it could be prone to lack of diversity. Is there a diversity evaluation result?

**Limitations:**

yes

**Strengths And Weaknesses:**

Strengths:
1. The experimental design is comprehensive and the results are pretty strong, especially on the comparison of sample efficiency. This could provide insights on the data curation for alignment of text-to-image models.
2. The paper is overall well-written.

Weaknesses:
1. The ablation study shows that a significant portion of the initial performance jump comes from Stage 1 (Distribution-DRO) which is an existing method. Then the contribution is mainly from the data curation pipeline, which could be highly dependent on the base model, the inpaint model, and the evaluation metrics.
2. The quality of the negative samples is entirely dependent on the flaws of the SD v1.5 inpainting model; it is unclear how the method would perform if the inpainter were too good to provide mistakes.

---

> ### Author Rebuttal · Authors · 2026-03-31
>
> > **Q1-1: Much of the initial gain comes from Stage 1**
>
> We respectfully clarify that while Stage 1 adopts Diffusion-DRO, our contribution is not merely algorithmic. The key insight is data-centric: we show that real images can serve as effective preference supervision without human annotation, and we design a two-stage framework to make this feasible and effective.
>
> Stage 1 provides distribution alignment as initialization for Stage 2 preference learning; neither stage alone is sufficient. Stage 1 alone yields compromised gains; for example, HPSv3 improves from 6.83 to 7.33 when both stages are applied, whereas Stage 2 alone only gives marginal improvements, with HPSv3 reaching 6.11. The performance gain arises from their interaction rather than from either component in isolation.
>
> > **Q1-2: Pipeline dependency**
>
> We clarify that our data curation pipeline is designed to improve the quality and reliability of preference signals, rather than to depend on a specific base model, inpainting model, or evaluation metric.
>
> - Base Model. We test a diffusion-based model (SD-1.5) and a flow-matching model (SD-3.5-M), with consistent improvements observed across both architectures
> - Inpainting Model. We replace SD-v1.5-Inpainting with stronger alternatives such as PixArt-α and SD-3.5-M, and all variants consistently gain improvements. We further explore constructing negatives via text-to-image generation as a different perturbation strategy, which remains effective. This suggests that the pipeline does not rely on a particular inpainting model or specific artifacts.
> |Model|PickScore|ImgRwd|UniRwd|HPSv3|DeQA|Aes.|
> |---|:---:|:---:|:---:|:---:|:---:|:---:|
> |SD-1.5|20.65|0.16|2.92|5.98|3.70|5.48|
> |Diffusion-DPO|21.03|0.33|3.03|6.80|3.78|5.59|
> |Ours (Text-to-image generation)|21.05|0.35|3.04|7.09|4.00|5.65|
> |Ours (SD-v1.5-Inpainting)|21.04|0.38|3.06|7.33|3.96|5.64|
> |Ours (PixArt-α Inpainting)|21.05|0.36|3.04|7.16|3.98|5.61|
> |Ours (SD-3.5-M Inpainting)|20.96|0.35|3.04|7.20|3.89|5.62|
>
> - Evaluation Metrics. We observe consistent improvements across six automatic metrics covering preference, perceptual quality, and aesthetics, and the same trend is supported by human evaluation, where our method achieves 63.33% win rate over SD-3.5-M and 56.67% over Diffusion-DPO. Together, these results suggest that the observed gains are not driven by metric-specific bias.
>
> > **Q2-1: Negative sample quality**
>
> We clarify that our method does not rely on imperfections of a specific inpainting model, but on the ability to generate controlled localized variations for contrastive comparison.
>
> - First, our data curation pipeline explicitly removes weak or ambiguous negatives, ensuring that only informative contrastive pairs are retained.
> - Second, replacing SD-v1.5-Inpainting with stronger models such as PixArt-α and SD-3.5-M, or using text-to-image generation instead, yields similar improvements, indicating that the method is not tied to a specific inpainting model.
>
> > **Q2-2: Inpainter is too good**
>
> Insightful point. If the regenerated regions were indistinguishable from the original, the pairs would indeed provide little contrastive signal. However, our method does not rely on inpainting itself, but on generating contrastive variations. Inpainting can therefore be replaced by other perturbation mechanisms, such as text-to-image generation, which remains effective. In practice, stronger generators reduce obvious artifacts but still introduce subtle differences, yielding more fine-grained supervision rather than eliminating the learning signal.
>
> > **Q3: Human evaluation**
>
> We conducted a user study (18 participants) on our fine-tuned SD-3.5-M model. Prompts were sampled from HPDv2 following the protocol of Diffusion-DRO: 60 prompts spanning four categories (Anime, Concept-Art, Paintings, Photo; 15 each). Our method achieved a win rate of 63.33% over the SD-3.5-M baseline and 56.67% over Diffusion-DPO, further confirming that real-data-based supervision provides effective guidance for aligning diffusion models.
> ||Win (%)|Tie (%)|Lose (%)|
> |---|:---:|:---:|:---:|
> |Vs. SD-3.5-M|63.33|16.67|20.00|
> |Vs. Diffusion-DPO|56.67|10.00|33.33|
>
> > **Q4: Data diversity**
>
> Our approach emphasizes data quality over quantity, and Fig. 7 shows that increasing the dataset size from 256 to 512 improves performance, while further scaling yields diminishing returns, indicating that a relatively small but well-curated set is sufficient for effective alignment. We further discuss two complementary perspectives:
>
> - Data diversity: We classify the 512 curated reference images using Qwen3.5-27B into the 12 categories defined in HPDv3, and find that all 12 categories are covered, confirming broad semantic diversity.
>
> - Model output diversity: Fig. 2 shows that our method improves human preference while preserving output diversity, maintaining balanced performance across attributes such as realism, texture, and stylization.

---

> > ### Author Rebuttal · Reviewer_K2hq · 2026-04-03
> >
> > Thank the authors for the response. I appreciate the results from different inpainting models and human eval and strongly encourage to add those to the main draft. I will adjust my score accordingly.

---

> > > ### Author Response · Authors · 2026-04-04
> > >
> > > Dear Reviewer K2hq,
> > >
> > > Thank you for your positive feedback and thoughtful follow-up. We are glad that our response has helped address your concerns. We appreciate your insightful suggestions and will incorporate the clarifications from the rebuttal in the revision.
> > >
> > > Best,
> > >
> > > The Authors

---

### Decision · Program_Chairs · 2026-04-30

**Decision:**

Accept (regular)

**Comment:**

This paper investigates the use of curated real images as reference points for diffusion model preference alignment, proposing a two-stage pipeline for distribution and preference learning. The reviewers found the method's data efficiency quite interesting, as it remains competitive with Diffusion-DPO while using significantly fewer preference pairs. During the rebuttal, the authors successfully addressed primary concerns regarding human evaluation and generalization by providing user study results and evidence from stylized domains like anime and concept art. Reviewer X4Rx maintained a weak reject due to concerns about baseline fairness, but didn't engage with the authors during the discussion phase. Despite that, the other three reviewers (K2hq, 4Fzq, Q2fi) were satisfied by the newly added robustness and scaling experiments. The consensus highlights the work as a well-motivated and practically effective alternative to synthetic preference pairs. Given the strong empirical results and the satisfaction of the majority of the committee, the submission is recommended for acceptance.